# TALKING VEHICLES: COOPERATIVE DRIVING VIA NATURAL LANGUAGE

## ABSTRACT

Using natural language as a vehicle-to-vehicle (V2V) communication protocol offers the potential for autonomous vehicles to drive cooperatively not only with each other but also with human drivers. Simple and effective messages for sharing critical observations or negotiating plans to achieve coordination could improve traffic safety and efficiency compared to methods without communication. In this work, we propose a suite of traffic tasks in vehicle-to-vehicle autonomous driving where vehicles in a traffic scenario need to communicate in natural language to facilitate coordination in order to avoid an imminent collision and/or support efficient traffic flow, which we model as a general-sum partially observable stochastic game. To this end, this paper introduces a novel method, LLM+DEBRIEF, to learn a message generation and high-level command policy for autonomous vehicles through multi-agent discussion. To evaluate our method, we developed a gym-like simulation environment that contains a range of accident-prone driving scenarios that could be alleviated by communication. Our experimental results demonstrate that our method is more effective at generating meaningful and human-understandable natural language messages to facilitate cooperation and coordination than untrained LLMs. Our anonymous code and demo videos are available at https://anonymous.4open.science/r/talking-vehicles.

## 1 INTRODUCTION

State-of-the-art autonomous driving policies are commonly designed from the perspective of a single agent's sensors. Therefore, to enhance safety, it is essential to account for multi-agent interactions. Among the strategies employed is incorporating motion prediction or intention inference of other traffic participants in decision-making. Some models independently forecast the future movements of these traffic participants (Wu et al., 2023), whereas others account for mutual interactions among autonomous vehicles and other traffic agents (Seff et al., 2023). However, the challenge lies in the substantial uncertainty in predicting other drivers' intentions, often resulting in excessively cautious driving strategies (Rhinehart et al., 2021).

Given its cooperative nature, the safe driving problem could be largely simplified by enabling vehicles to communicate their intentions and observations with each other. The concepts of vehicle-to-vehicle (V2V) and vehicle-to-infrastructure (V2I) communication have recently emerged as a promising approach for multi-vehicle cooperation, garnering considerable research interest (Wang et al., 2020; Cui et al., 2022; Xu et al., 2022a;b). While the focus of V2V communication has predominantly been on cooperative perception rather than cooperative control, it often employs modalities (latent representations, LiDAR points, locations of objects, etc.,) that are not intuitively nor easily understood by humans, thus requiring that all the participating vehicles be autonomous and share the same protocol. Natural language, as a refined and highly adaptable form of human communication, offers the potential for human drivers to also participate in this cooperation. If vehicles could "speak" in human language, it would pave the way for developing technologies that facilitate communication between autonomous vehicles and human drivers, enhancing cooperation and understanding in mixed-autonomy traffic environments. Even in fully autonomous settings, a natural language interface could offer flexibility for agents to explain their decisions, talk

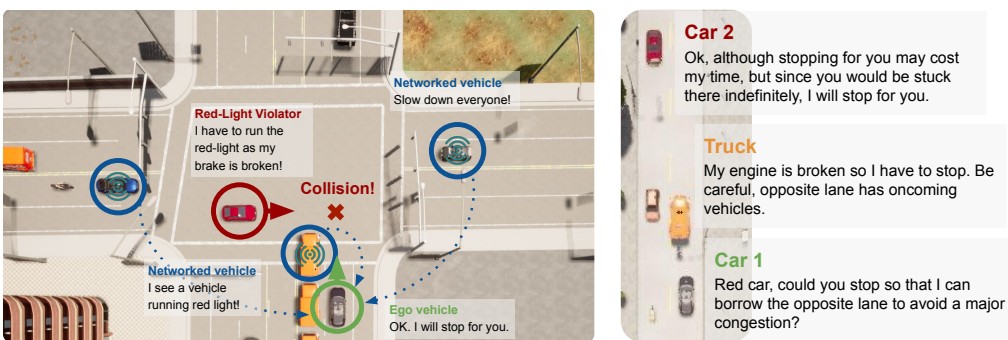

Figure 1: **Left:** A red-light violation scenario where there is a car running the red light. **Right:** An overtake scenario where a truck is broken and stopped on a two-lane two-way road.

to other autonomous vehicles with proper translation, quickly adapt to different traffic regulations, and negotiate plans.

As a concrete example of the power of using language in V2V communication, consider the following scenario, as illustrated in Figure 1. A truck has broken down and stopped on a two-way road divided by a yellow dashed line. The vehicles trailing the truck cannot see whether the lane going in the opposite direction is free from oncoming cars, which could allow them to use that lane to safely move around the truck to circumvent the growing traffic jam. If the truck is capable of communication, it could relay a message such as, "The opposite lane is clear, you may safely perform a lane change to pass me." Alternatively, in busier times, the queued vehicles could initiate a dialogue with oncoming traffic, proposing, "Can we take turns using the lane to avoid major congestion?" Through simple but effective communication about their intentions or observations, a mixture of human-driven and autonomous vehicles could significantly enhance traffic flow and overall efficiency.

In this work, we first introduce the problem of *talking vehicles*. In this problem setting, a scenario in autonomous driving is formulated as a multi-agent partially observable and general-sum game wherein each traffic participant is pursuing a cooperative goal modulated by individual preferences[1]. When conflicts arise or unexpected events occur, vehicles have the opportunity to generate and broadcast messages that contain observations of abnormal events or negotiations to cooperatively reach their goals. The vehicles can then make decisions according to the received messages.

Enabling vehicles to **"talk"** to each other with intentions and conveying helpful information in natural language presents significant challenges. There are studies like Dolphins (Ma et al., 2023) and LINGO-1 (Wayve, 2023) that have trained Visual-Language-Action models to both make driving decisions and articulate their reasoning to humans. However, training such models requires extensive data. At the time of writing this paper, only a limited number of datasets exist that provide language commentary data for single-agent driving scenarios (Kim et al., 2018; 2019; Qian et al., 2023; Sima et al., 2023) or in-vehicle communication with human driver (Deruyttere et al., 2022). To the best of our knowledge, datasets featuring natural language data for **inter-vehicle** communication are not yet available.

On the other hand, multi-agent **"self-play"** learning in a high-fidelity simulator requires no real-world data and allows for closed-loop learning and evaluation. With recent advances in large language models (LLMs), there has been a surge of interest in applying LLMs to multi-agent games (Bakhtin et al., 2022; Xu et al., 2023a; Light et al., 2023). However, the synergy between LLMs and multi-agent games remains unexplored in the context of autonomous driving. Existing efforts in autonomous driving mainly focus on leveraging LLMs for decision-making in a *single-agent* setting (Mao et al., 2023a; Shao et al., 2023; Ma et al., 2023).

---

[1]Generally, we assume all cars seek a smooth traffic flow. Meanwhile, each car prefers to reach its destination as quickly as possible. But their goals may conflict with each other's.

To enable autonomous agents to learn what to say to cooperate from interactions with each other while still constrain the messages in the natural language space, we propose a novel method, LLM+DEBRIEF, as an initial attempt to use LLMs to help solve the cooperative driving problem in a V2V communication framework. This method optimizes the in-context knowledge of a language model as the message generation guideline through a turn-based post-episode discussion(debriefing) and leverages LLMs to analyze the dialogue history among neighboring vehicles to generate general driving commands and messages to send. Finally, an atomic controller executes the control according to the updated instructions. To test our method and provide a research test bed for the community, we build a simulation framework containing an array of interesting multi-agent driving scenarios that support communications in natural languages. Our experimental results show an improvement in driving safety and efficiency of our method compared to the methods without communication.

In summary, this paper makes the following contributions:

1. We present the task of cooperative driving facilitated by vehicle-to-vehicle communication using natural language.

2. We contribute a multi-agent simulation framework to realistically model vehicle-to-vehicle communication through natural language, featuring diverse scenarios that capture a range of traffic conditions and interactive challenges.

3. We propose a novel method that empowers autonomous vehicles to dynamically generate and integrate natural language messages, facilitating inter-vehicle cooperation and leveraging multi-agent debriefing for improved coordination after interactions.

4. We evaluate our methodology within this simulation framework, and find that our method is able to generate meaningful messages and improve traffic safety as well as efficiency compared to alternative approaches.

## 2 RELATED WORK

**Vehicle-to-vehicle Communication.**    Vehicle-to-Vehicle (V2V) or Vehicle-to-everything (V2X) communication offers the potential to effectively facilitate multi-vehicle cooperation, improving the safety and reliability of autonomous vehicles in urban driving scenarios. Existing research predominantly concentrates on cooperative perception data sets (Yu et al., 2022; Xu et al., 2022b; Li et al., 2022) and tasks like cooperative detection and prediction (Wang et al., 2020; Chen et al., 2019; Xu et al., 2022a), leveraging sensor data from cameras, LiDAR, and other resources. The message aggregation strategies include early fusion (Qiu et al., 2022), late fusion, and intermediate fusion (Wang et al., 2020). Considering the limited V2V bandwidth, efforts were made to reduce the message size (Hu et al., 2022). Aoki et al. (2020) developed a reinforcement learning method for selecting what information to be transmitted in cooperative perception. Although Cui et al. (2022) developed end-to-end driving policies and derived the critical information from expert supervision, the cooperation is still completed at the perception level. In contrast, our work focuses on the message in the natural language space to enhance both cooperative perception and formulation of cooperative driving strategies. Real-world communication often suffers from caveats in the communication mechanisms, including packet loss, latencies (Lei et al., 2022), and localization errors. Although some works consider adversarial attacks (Tu et al., 2021) in V2V communication, we assume all vehicles are cooperative in this work.

**Multi-Agent Social Interactions with LLMs.**    Large Language Models (LLMs) show promising communication and reasoning capabilities, suggesting their potential in multi-agent interaction scenarios. For instance, Generative Agents (Park et al., 2023) represents an early attempt at employing LLM agents for free-form chatting, demonstrating the believable behaviors of LLM agents in spreading information. However, this study did not evaluate the LLMs' capabilities in planning or solving multi-agent tasks. Cicero (Bakhtin et al., 2022), on the other hand, finetunes a language model to imitate human behaviors from a dataset to generate truthful messages in the game of Diplomacy, which is mixed-motive and requires communication in natural language. They train reinforcement learning policies to analyze the dialogue select actions and generate deceptive messages through value filtering. Recent

works on communication games Werewolf (Xu et al., 2023a) and Avalon (Light et al., 2023) demonstrate that LLMs can achieve impressive results in multi-agent settings. This success is particularly notable when LLMs are combined with reinforcement learning or fine-tuning. Shi et al. (2023) find that LLMs can be applied to achieve Ad Hoc teamwork in the Avalon game. Recently, a generative agent-based social simulator (Vezhnevets et al., 2023) has been proposed to serve as a test bed for multi-agent LLM interactions as well.

**LLMs for Autonomous Driving.** LLMs have shown great potential in solving various autonomous driving tasks. In particular, they are promising in tackling corner cases (Wen et al., 2023b) due to their reasoning ability and the common-sense knowledge embedded, yielding a more generalizable autonomous driving stack. Recent studies have explored various approaches to tailor state-of-the-art LLMs for driving. Similar to other embodied tasks (Driess et al., 2023; Brohan et al., 2023), a foundational challenge lies in grounding LLMs in the real world—the LLMs need to perceive and understand the traffic scenarios. A straightforward approach is to obtain the observations from oracle perception models (Mao et al., 2023b) and convert them to textual descriptions (Mao et al., 2023a; Sha et al., 2023; Jin et al., 2023; Cui et al., 2023). Some other studies tackled this challenge by introducing Visual Language Models (VLMs), which are adapted to driving domains through in-context instruction tuning (Ma et al., 2023) or fine-tuning (Wayve, 2023; Xu et al., 2023b; Ding et al., 2023; Yang et al., 2023). To enhance LLMs' reasoning ability, prior works have investigated incorporating handcrafted guidance and examples in the prompts (Sha et al., 2023; Jin et al., 2023; Cui et al., 2023), structuring the reasoning procedure (Mao et al., 2023b; Sima et al., 2023), and fine-tuning the models on driving datasets. Notably, fine-tuning LLMs and VLMs requires an extensive amount of driving data with language labels. While a limited number of such datasets are available (Kim et al., 2018; 2019; Malla et al., 2023), they were mostly created in the pre-LLM era and, thus, are not designed for LLM fine-tuning. While several works have attempted to adapt existing language-driving datasets for LLM fine-tuning (Ding et al., 2023; Xu et al., 2023b; Ma et al., 2023), growing attention has been drawn to directly augment large-scale multimodal driving datasets, such as nuScenes (Caesar et al., 2020), Waymo (Sun et al., 2020), and ONCE (Mao et al., 2021)), with language labels (Qian et al., 2023; Shao et al., 2023; Sima et al., 2023; Nie et al., 2023). Note that existing models were predominantly evaluated in *open-loop* fashions, except for Shao et al. (2023); Sha et al. (2023); Jin et al. (2023). The open-loop evaluation results may not effectively imply the models' closed-loop performance after deployment. In contrast, similar to Surrealdriver (Jin et al., 2023), we conduct closed-loop tests of the proposed multi-agent communication and control framework in CARLA (Dosovitskiy et al., 2017). More importantly, none of the existing works have explored LLMs in a multi-agent setting with V2V communication as we did. LanguageMPC (Sha et al., 2023) was demonstrated in a multi-agent scenario, yet its controller is centralized.

## 3 PROBLEM DEFINITION

We frame the problem of **Talking Vehicles** as a general-sum partially observable stochastic game (POSG), focusing on optimizing the social welfare of a *focal population* ($\mathcal{F}$) (Agapiou et al., 2022) — defined as the cumulative reward of all agents of interest — as the primary objective. This problem is a general-sum game because it includes scenarios that are not entirely cooperative; conflicts of interest may exist and some agents may have to sacrifice their individual interests for the overall benefits. Each agent's observation space is limited to a partial view of the full state, and agents make decisions in a decentralized manner based on ego partial observations and received messages from other agents. Within this problem, the action space for each agent has two main components: 1. the generation of messages, and 2. the control of the vehicle. In this work, the message generation space is constrained within natural language.

A POSG can be described by the tuple $\langle \mathcal{I}, \mathcal{S}, \{\mathcal{O}_i\}, \{\mathcal{A}_i\}, \mathcal{P}, \{\mathcal{R}_i\}, \gamma \rangle$, where $\mathcal{I} = \{1, 2, ..., N\}$ refers to the identities of actionable agents; $\mathcal{S}$ is the state space comprehensively describing the environment; $\mathcal{O}_i$ is the observation space of agent $i$'s state; $\mathcal{A}_i$ is the action space of agent $i$; $\mathcal{P}$ represents the state transition function $\mathcal{S} \times \mathcal{A}_1 \times \mathcal{A}_2 \times ... \times \mathcal{A}_N \to \mathcal{S}$; $\mathcal{R}_i$ is the reward function of agent $i$; Finally, $\gamma$ is the discount factor.

In this problem, the goal for each agent $i \in \mathcal{I}$ is to optimize a policy $\pi_i$ to maximize the expected sum of all the agents' returns in the focal population $\mathcal{F} \subseteq \mathcal{I}$:

$$\max_{\pi_1, \pi_2, \ldots, \pi_N} \quad \mathbb{E}_{\pi_1, \pi_2, \ldots, \pi_N} \left[ \sum_{i \in \mathcal{F}} R_i \right] \tag{1}$$

The policy $\pi_i(O_i, \{M_j\}_{j \in \mathcal{I}}) \to \mathcal{A}_i$ maps the observation of agent $i$ and received messages $\{M_j\}_{j \in \mathcal{I}}$ to its action space $\mathcal{A}_i = \langle \mathcal{M}_i, \mathcal{C}_i \rangle$, where $\mathcal{M}_i$ is the message generation space, and $\mathcal{C}_i$ is the motion command space which includes a sequence of low-level control with dimensions for throttle, brake, and steering inputs. The generated message $M_i$ by agent $i$ at time step $t$ is broadcast to all other connected agents within a specific communication radius at the next time step $t + 1$.

In summary, the essence of the *talking vehicles* problem is to enable each agent to derive effective control-communication strategies from its observations and the messages it receives. These strategies, coupled with appropriate vehicle control actions, aim to achieve coordinated driving behavior.

This problem presents the following technical challenges to the machine-learning community:

1. How can learned agents understand the situation and **generate** meaningful messages to help others perceive the environment or potentially negotiate about motion plans;

2. How can learned agents **comprehend** the received messages and **incorporate** them into high-level **cooperative** driving decisions.

In this work, we assume that the agents could communicate truthfully, meaning they accurately convey their real intentions and follow through on their stated decisions. However, since they act simultaneously, other agents will only be able to process the received message in the next decision step. Additionally, we assume that all agents aim to cooperate with the focal population and do not send deceptive messages or act aggressively to sabotage the driving goals of others intentionally. Exploring scenarios where these assumptions are lifted could be an interesting direction for future work.

## 4 ENVIRONMENT

To provide concrete and typical driving scenarios that expose the *talking vehicles* challenge, we have developed a simulation environment, **TalkingVehiclesGym**, which is a multi-agent gymnasium environment for closed-loop evaluation of urban driving policies. TalkingVehiclesGym provides a flexible configuration of heterogeneous agents (such as language agents, sensory agents, human agents, behavior agents, etc.) and policies in the environment. This framework also enables **in-episode** communication capabilities of agents using a realistically simulated communication protocol MQTT and the dynamic simulation is built upon CARLA (Dosovitskiy et al., 2017), a high-fidelity urban driving simulator.

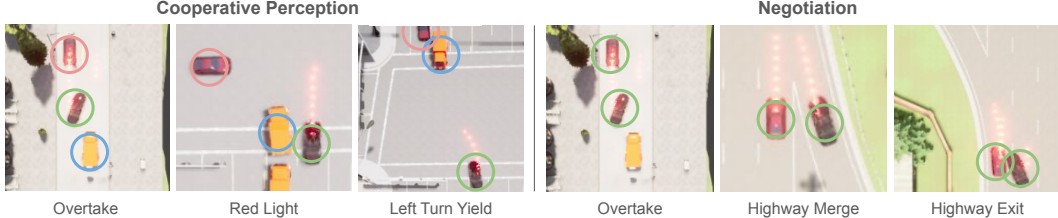

Figure 2: *Overview of Test Scenarios.* Agent roles are marked with circles with different colors. Red: Potential Colliders; Green: Focal Agents, agents having both driving control and communication capabilities; Blue: Other Cooperative Agents, agents that can communicate to help the focal agents. Detailed descriptions of environment dynamics are in Table 4, Appendix B.

**Scenarios.** TalkingVehiclesGym has been set up with several accident-prone scenarios where multi-agent communication could be advantageous (Figure 2). Scenarios labeled with `Cooperative Perception` represent opportunities for agents to benefit from shared information about areas outside their immediate line of sight. On the other hand, scenarios tagged with `Negotiation` are designed to demonstrate the advantages of agents discussing and reconciling their plans when conflicts arise. For a detailed description of the design structure of TalkingVehiclesGym, and scenario descriptions (Table 4), please refer to Appendix B.

**Perception.** A wide array of sensors are also available for models that handle corresponding modalities. To simplify environmental perception for language-only models, TalkingVehiclesGym is equipped with a rule-based **partially-observable** captioner that translates the observation into text for an agent but preserves the partial observability of line-of-sight sensors (Example text observation description in Appendix C).

**Atomic Actions.** Since LLMs take considerable time to generate reasoning and decisions, and are not highly accurate with numerical tasks, it is currently impractical to use them for controlling low-level vehicular motions. Instead, this paper focuses on high-level decision-making with natural language communications. Our multi-agent communication and simulation framework allows models to consider only high-level plans, abstracting trajectories into a sequence of atomic actions including `go, stop, slow down, speed up, change to the left lane, change to the right lane`.

## 5 METHOD

The core technical problem that we address is to let agents communicate purposely to facilitate cooperation and act correspondingly using human language as the medium. Training language policies using gradient-based methods to perform particular communication tasks by self-play is known to generate non-human-comprehensible artificial languages. To establish an initial solution to the *talking vehicles* problem, we start with an agentic framework that prompts large language models as a foundational prior for autonomous agents to engage in human-like communication, regularizing the message within natural language space, allowing agents to interpret messages and make informed driving decisions. A key challenge of using LLMs lies in that they are not specifically trained for driving tasks. To overcome this limitation, we introduce LLM+DEBRIEF (Algorithm 1), a novel agentic framework built upon feedback loops that allow LLMs agents to iteratively refine their communication and motion policy through trial-and-error interactions with confederate agents. Additionaly, inspired by how humans reflect and debrief after a Hanabi game, we enable agents to discuss cooperative strategies after each interaction episode. As illustrated in Figure 3, our method consists of three core components: **In-episode communication, Chain-of-Thought Reasoning**, and **Post-Episode Debriefing**.

**In-episode Communication.** Each driving agent is equipped with a transceiver module that enables real-time communication during episodes. Agents broadcast and receive structured messages by subscribing to topic-specific communication channels. Each message follows a predefined format containing key details such as content, timestamp, sender ID, and sender location. This structured approach ensures that messages are contextually relevant and easily interpretable within multi-agent communication. Received messages are stored in a buffer, and recent message dialogs are incorporated into the agent's observations for decision-making.

**Chain-of-Thought Reasoning.** Ye et al. (2024) observed that current LLMs can make irreversible mistakes when computing variables without sufficient context. To address this, we prompt the LLM to first reason about the environment based on its task, observations, received messages, cooperative strategy, and accumulated knowledge before making decisions. After reasoning, the LLM generates actions in a structured JSON format with keys: {`"command"`, `"message"`}. These outputs are then translated into vehicle controls and communication messages to publish. All observations, commands, messages, and reasoning are stored in a **replay buffer** for further learning and refinement.

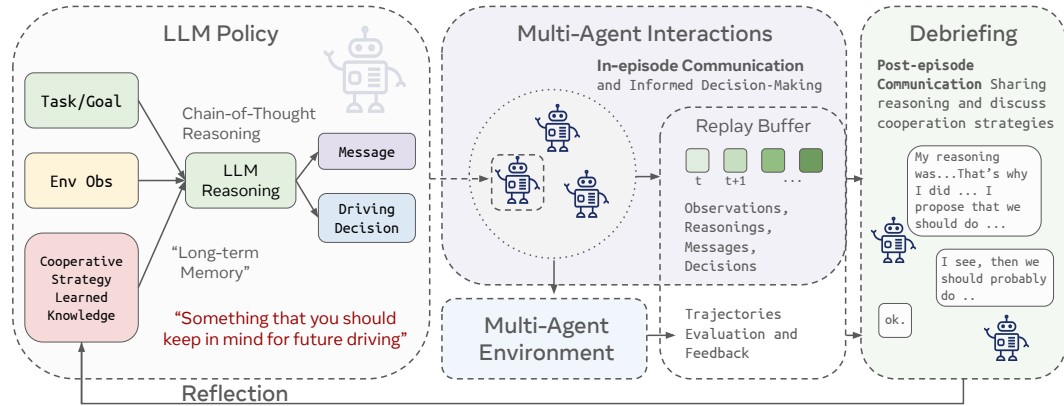

Figure 3: Method. A LLM+DEBRIEF policy is provided a task to complete for each scenario, and the environment will provide the text observation of the surroundings and message dialogues. Along with the previous learned knowledge, the policy first performs chain-of-thought reasoning about all the inputs, generates messages to others, and drives decisions based on observation and in-episode communication. Then, the agents within the multi-agent environment will make the decisions based on their corresponding policies. After each episode, agents receive feedback from the environment evaluation containing information like timeout, success, or collision information. This feedback, along with the {observation, reasoning, message, commands, and others' reactions}, are stored in a replay buffer for future learning. During the debriefing phase, agents are able to revise their strategy and knowledge about decision-making by learning others' reasoning during and after episodes. Such knowledge is then stored as a long-term knowledge for future decision-making.

**Post-Episode Debriefing.** When an episode concludes, the environment evaluates each agent's performance and provides rich semantic feedback, such as "Vehicle 109 collided with Vehicle 110 in 2 seconds" or "Vehicle 109 stagnated too long to complete its task." Each learning data point in the replay buffer is **retrospectively labeled** with additional information, including other agents' responses, collision details, and stagnation details.

Before engaging in the post-episode discussion(**debriefing**), each learning agent replays and reflects the past experience by sampling a **batch** of learning data from its own replay buffer. The sampling process heuristically assigns higher probabilities to data that are pre-collision, slowing down agents in stagnation, and involving intensive interactions; these samples will serve as the context for analysis and strategy formulation.

The debriefing is conducted in a **turn-based** manner over $N$ rounds, centered around improving cooperation in future interactions. In each discussion session, a discussion host randomly decides the speaking order. The agent selected to speak first gets the opportunity to propose a cooperative strategy. Other agents then take turns to express their opinions or thoughts on the proposed strategy.

After debriefing, agents gain a clearer understanding of one another and summarize the discussion to form cooperative strategies and individual knowledge as in-context guidelines for future driving tasks. Chain-of-thought reasoning is applied during debriefing to reinforce decision-making processes. This structured post-episode analysis resembles the Centralized Training Decentralized Execution (CTDE) framework commonly used in multi-agent learning.

## 5.1 IMPLEMENTATION DETAILS

We employ `Llama-3-8B-Instruct` (Dubey et al., 2024) at a temperature of 0.2 for the agent framework, deciding and collecting experiences every 0.5 seconds (10 simulation frames). The received message dialog is maintained within 2 seconds of the message age during the episode. The debriefing process happens after each episode for 30 episodes and spans 1 round of discussion, followed by a final round of individual reflection steps to summarize the discussion. A batch size of 2 is used to sample transition data from the trajectory. Additional details including decision latencies can be found in Appendix A.

## 6 EXPERIMENTS

In this section, we aim to verify the following hypotheses with empirical experiments:

1. LLM agents can perform better with communication than without; LLM agents can effectively respond to natural language messages without training and cooperate with each other through communication to improve their performance;

2. LLM agents can autonomously improve themselves in the collaboration tasks through decentralized reflection without human intervention;

3. Debriefing can further enhance LLM learning from interactions more effectively than decentralized reflection alone;

**Metrics.** In each scenario, we define a focal population whose movements we are interested in and can control to accomplish a specific task within a certain time limit. Evaluation metrics are then established based on the performance of this **focal** population over 30 evaluation episodes. We utilize three key metrics: the average total reward (**R**) accrued by the focal population, the average number of collisions per episode (**CR**) of the population, and the average episodic success rate (**SR**) normalized by the size of the population. Success is defined as reaching target locations within a designated time frame without collision. An agent who successfully completes the task earns a reward of $+1$. Conversely, collision incurs a penalty of $-1$ for each agent involved in the collision while remaining stagnant at any point until timeout results in a reward of 0 because, although not ideal, conservative policies are at least safe.

**Baselines.** We established several baselines and scenarios to evaluate our hypothesis. These baselines include: (1) an untrained LLM, (2) an LLM trained with decentralized reflection that updates in-context knowledge (LLM+Reflection), (3) an LLM that corrects past actions via decentralized reflection, storing these corrections in a vector-based, retrievable memory and uses few-shot retrieval augmented generation (LLM+Reflection+RAG), and (4) an LLM trained with debrief discussions as outlined in Section 5 (LLM+Debrief). The retrieval augmented method without communication adapts DiLU (Wen et al., 2023a), a non-communicating single-agent LLM-based approach that drives via reflection, to our environment. The multi-agent communication extension of DiLU, AgentsCoDriver (Hu et al., 2024), resembles the Reflection+RAG (Comm) method, but they do not actively optimize the messages. For a fair comparison across DiLU, AgentsCoDriver, and other baselines, we do not initialize the knowledge with human data, nor is there human involvement during the learning process. Additionally, we include Coopernaut (Cui et al., 2022), a LiDAR-based cooperative driving method, as a reference for cooperative perception. Note that since Coopernaut relies on intermediate sensor data representations rather than natural language communication, its results are not directly comparable to the other methods being compared.

**Experiment Setup.** For each baseline[2], we consider two settings labeled as "Silent" and "Comm", respectively. In the "Silent" setting, the method operates without communication, where policies focus solely on controlling the vehicle without generating messages. In contrast, the "Comm" setting allows the method to either generate messages alone or both messages and driving commands. For each LLM-based learning method, we train the models for up to 30 episodes per scenario, with early stopping if the scenario is solved, indicated by 10 consecutive successful episodes. After training, we evaluate each method over 30 episodes and report the average performance across these evaluations.

### 6.1 QUANTITATIVE RESULT

**LLMs can facilitate cooperation through language communication in zero-shot.**
Table 1 presents our evaluation of LLM agents without training across all scenarios under both the "Comm" and "Silent" settings. We observe that, even without learning, LLMs are able

---

[2]Except for LLM+Debrief, which is only tested under the "Comm" setting since it is particularly designed for improving multi-agent communication.

Table 1: Experiment Results for Communication vs Silent Agents. We evaluate on adversarial cases where being aggressive or conservative will result in failure.

| Scenario / Method | | | Overtake (Perception) | | | Red Light | | | Left Turn | | |
|---|---|---|---|---|---|---|---|---|---|---|---|
| Name | LLM | Comm | R ↑ | CR ↓ | SR ↑ | R ↑ | CR ↓ | SR ↑ | R ↑ | CR ↓ | SR ↑ |
| Untrained (Silent) | Yes | No | 0.00 | 0.00 | 0.0 | -0.60 | 0.80 | 20.0 | 0.20 | 0.33 | 53.3 |
| Untrained (Comm) | Yes | Yes | -0.63 | 0.66 | 33.3 | 0.80 | 0.07 | 86.7 | 0.20 | 0.33 | 56.7 |
| Coopernaut | No | Yes | 1.00 | 0.00 | 100.0 | 0.97 | 0.00 | 96.7 | 0.93 | 0.03 | 96.7 |

| Scenario / Method | | | Overtake (Negotiation) | | | Highway Exit | | | Highway Merge | | |
|---|---|---|---|---|---|---|---|---|---|---|---|
| Untrained (Silent) | Yes | No | 0.50 | 0.67 | 3.3 | -0.93 | 1.5 | 20.0 | -1.26 | 1.63 | 18.5 |
| Untrained (Comm) | Yes | Yes | 0.50 | 0.70 | 53.3 | -0.73 | 0.6 | 33.3 | -0.10 | 1.03 | 45.5 |

to leverage communication to foster some levels of cooperation in most scenarios, indicated by higher success rates when addressing conflict or partial observation challenges. However, there remains substantial room for improvement. Interestingly, while communication enables cooperation, it also tends to increase the frequency of collisions compared to the Silent setting. We hypothesise that it is because LLMs become overly confident in their perceptions or behave more aggressively in driving tasks. In contrast, without communication, LLMs often adopt overly conservative policies, particularly in scenarios like Overtake (Perception). The typical message length generated by LLMs ranges from 0 to 50 words, requiring less than 0.01 Mbps, a stark contrast to the 5.1 Mbps reported in Coopernaut Cui et al. (2022), highlighting the efficiency of using natural language as the communication protocol, especially in negotiation tasks.

Table 2: Experiment Results for Improvement Methods.

| Scenario / Method | | | Red Light | | | Highway Merge | | |
|---|---|---|---|---|---|---|---|---|
| Name | LLM | Comm | R ↑ | CR ↓ | SR ↑ | R ↑ | CR ↓ | SR ↑ |
| Untrained | Yes | No | -0.6 | 0.80 | 20.0 | -1.26 | 1.63 | 18.5 |
| +Reflection | Yes | No | -0.73 | 0.86 | 13.3 | -0.86 | 1.43 | 28.5 |
| +Reflection+RAG | Yes | No | -1.00 | 1.00 | 0.00 | -2.00 | 2.00 | 0.0 |
| Untrained | Yes | Yes | 0.80 | 0.07 | 86.7 | -0.10 | 1.03 | 45.5 |
| +Reflection | Yes | Yes | 0.70 | 0.13 | 83.3 | 0.20 | 0.87 | 50.0 |
| +Reflection+RAG | Yes | Yes | -0.93 | 0.96 | 3.3 | -2.00 | 2.00 | 0.0 |
| +Debrief | Yes | Yes | 0.80 | 0.07 | **90.0** | **0.40** | **0.57** | **51.5** |

**LLMs can be further improved through reflection and debriefing.** Table 2 evaluates different training methods in Red Light (Perception) and Highway Merge (Negotiation) scenarios. We found that LLMs' performance in negotiation tasks improves with reflection, but incorporating in-context knowledge updates and revising them with new experiences proves more reliable than LLMs' self-correcting actions without human oversight.

We hypothesize that the failure of Retrieval Augmented Generation (RAG) methods in our environment stems from the complexity of generating accurate messages—the search space is too large for language models to easily correct without additional validation. Improvements from reflection were less pronounced in perception tasks but significantly enhanced performance in negotiation tasks, where reflection benefited both silent and communication settings. The LLM+Debrief method achieved the best performance overall, underscoring the potential of collective discussion in improving cooperation.

## 6.2 QUALITATIVE ANALYSIS

While the main cooperation mode in perception tasks is sharing critical or abnormal traffic information (Appendix F.1, F.2, F.3), the cooperation mode in negotiation mainly lies in the argument on road priority Appendix F.4, F.5, F.6). Interestingly, we found that LLMs can form a convention through in-episode communication and demonstrate diverse conventions across episodes. For example, in the evaluation of the LLM+Debrief model, we found that

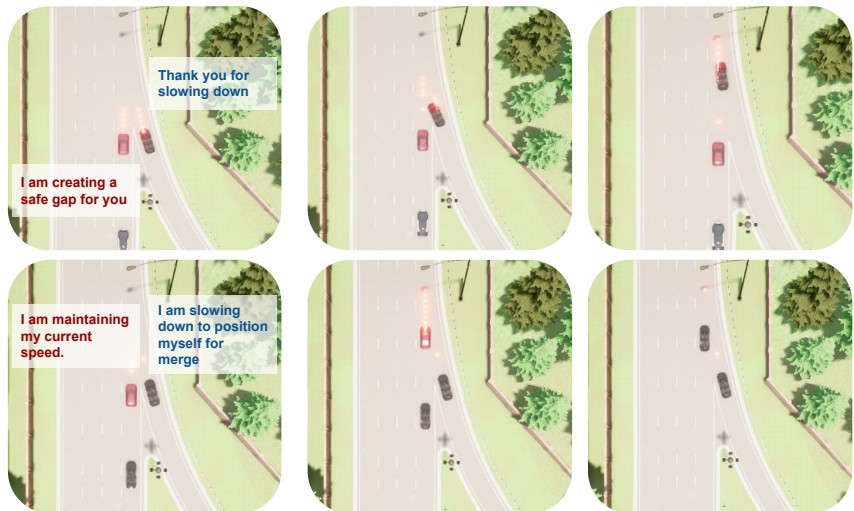

Figure 4: Diverse cooperation modes in the Highway Merge negotiation scenario. In this scenario, both vehicles aim to merge quickly as they are in a hurry. **Top**: The merging vehicle successfully negotiates for the highway vehicles to yield. **Bottom**: The merging vehicle opts to wait for the highway vehicle to pass before merging. We refer readers to the demo videos for detailed messages.

LLMs could gives way to vehicles on the main highway, while sometimes successfully convince the highway cars to slow down for them, demonstrated in Figure 4.

In Red Light Violation scenario, we compare the generated cooperative strategy and knowledge for future driving from reflection and debriefing. We found that the debriefed knowledge and cooperative strategies are more comprehensive and proactive than the decentralized reflection knowledge. Details in Appendix F.

## 7 Conclusion and Future Work

In summary, we identify a novel application domain for large language models in multi-agent learning systems, *talking vehicles*, where agents are required to send natural language messages and understand natural language messages to incorporate them into driving plans. Solutions to the *talking vehicles* problem have the potential to enable autonomous agents to facilitate cooperative perception and negotiation with human drivers. As a first attempt to solve the *talking vehicles* challenge, we propose a new method, LLM+Debrief, for generating messages and comprehending received messages. Our experiments show the effectiveness of LLM+Debrief quantitatively and qualitatively.

**Limitations and Future Work.** While we provide initial evidence of LLM+Debrief's potential in the *talking vehicles* problem, this research opens up several exciting future research areas for further exploration and development. First, the current LLM+Debrief framework takes text descriptions as observations, which relies on an idealized perception system. The TalkingVehiclesGym environment is able to provide multi-modal sensor observations. In future work, we are interested in developing a multi-modal extension of LLM+Debrief, which allows end-to-end perception and reasoning over the rich context information embedded in multi-modal observations. Second, this paper reports on a successful proof-of-concept, we are interested in scaling the evaluation benchmark and solution to more diverse traffic scenarios and operation conditions resembling real-world V2V communication, e.g., subject to time delays, adversarial attacks, and limited bandwidths. Last, our framework opens up the exciting potential to create a cooperative driving system for mixed-autonomy traffic scenarios. We are interested in realizing this potential, by studying the *talking vehicles* problem with human-in-the-loop experiments and exploring framework design to enable efficient communication between autonomous vehicles and human drivers. We refer readers to Appendix E for a full discussion of limitations and future work.

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

## A    Method

The Algorithm 1 implements LLM+Debrief, a centralized multi-agent learning framework that leverages communication and reflection using large language models (LLMs) to enhance coordination between agents in a simulated environment.

Messages exchanged between agents during the simulation are incorporated into their observations, allowing the agents to adapt their strategies continuously. After completing each episode, the environment provides feedback, which is used to label and process the experiences in the replay buffer for further learning. This feedback helps the agents refine their knowledge and improve their decision-making in subsequent episodes.

The debriefing process plays a crucial role in this algorithm. After each episode, agents engage in multiple rounds of debriefing, where they propose or revise cooperative strategies based on their experiences and interactions. The order of debriefing is randomized to simulate natural dialogues, enhancing the realism of the communication. Once the debriefing rounds are complete, the agents reflect individually, summarizing the discussions and updating their knowledge bases. This reflection step is critical for improving future performance, enabling agents to learn from successes and failures.

At the conclusion of the training process, the agents' knowledge and policies are updated, with the final policies from the last self-play iteration being used for further evaluations. The entire process is designed to improve the agents' ability to communicate effectively and make informed decisions in a multi-agent setting.

The agents use `Llama-3-8B-Instruct` to generate and interpret messages, with a temperature setting of 0.2 to ensure more deterministic outputs. The environment updates every 0.5 seconds (equivalent to 10 simulation frames), and the agents' messages are considered relevant for up to 2 seconds, ensuring timely and efficient communication. The experiments were conducted on two Nvidia A40 40GB GPUs, which were used to manage both the LLM-based policies and the simulation environment. This setup allowed the agents to run their LLM-based decision-making processes in parallel, enhancing the scalability of the system and enabling more efficient training.

Table 3 summarizes the average latencies and message sizes for each scenario under the communication setting, evaluated using Llama3-8B-Instruct on Nvidia A100 GPUs and Intel Gen 10 CPUs. The metrics include partial observable captioner latency (in seconds), reasoning latency (in seconds), decision latency (in seconds, excluding reasoning latency), and message size (in Mb). Data is aggregated over 10 episodes at each LLM decision step. Notably, GPT-4o online APIs demonstrate 2x faster generation speeds (16 seconds vs 8 seconds). Scenarios without communication exhibit slightly lower reasoning and decision latencies compared to those with communication (16 seconds vs 13 seconds), though the differences are within the same order of magnitude.

Table 3: Captioning, Reasoning, Decision Latency, Message Size using Llama3-8B-Instruct LLM Policy on Nvidia A100 GPUs.

| Scenario / Latencies | Overtake | Left Turn | Red Light | Overtake | Highway Merge | Highway Exit |
|---|---|---|---|---|---|---|
| Captioner Latency (s) | 0.022 | 0.023 | 0.025 | 0.022 | 0.017 | 0.016 |
| Reasoning Latency (s) | 18.32 | 16.89 | 16.93 | 12.57 | 18.10 | 18.48 |
| Decision Latency (s) | 2.83 | 2.25 | 2.37 | 1.56 | 1.57 | 1.60 |
| Message Size (Mb) | 0.0016 | 0.0013 | 0.0014 | 0.0014 | 0.0005 | 0.0005 |

---

**Algorithm 1** Multi-Agent Centralized Debrief Reflection with Communication

---

**Input:** Multi-agent Simulation Environment `env`, LLM agents$\{\pi_{i \in \mathcal{I}}\}$, Debriefing round $R$.

**Initialize:** Knowledge $\{K_{i \in \mathcal{I}}\}$, Replay Buffer *ReplayBuffer*

**for** $j$=1, 2, 3... // Training epoch **do**

  $\{\text{obs}_i\}$ = `env.reset`()

  **while** t<T // Time step **do**

    **for** i=1,..., N //Per agent, but execute in parallel **do**

      // Get CoT reasoning for each agent based on observation and knowledge

      reasoning$_i$ ←`agents.reason`(obs$_i$, $K_i$)

      // Get decisions for each agent based on observation and knowledge

      message$_i$, control$_i$ ←`agents.act`(obs, $K_i$, reasoning$_i$)

    **end for**

    // Step the environment with actions

    $\{\text{next\_obs}_i\}$ ←`env.step`($\{$message$_i$, control$_i\}$)

    // Store experience to the replay buffer

    *ReplayBuffer*`.add`(obs, next\_obs, reasonings, messages)

    // Message Dialog becomes part of the observation

    $\{\text{obs}_i\}$ ←$\{$next\_obs$_i\} \cup \{$message$_i\}$

  **end while**

  // Get episode feedback from the environment

  feedback ←`env.evaluate`()

  // Lable all the transition data in hindsight

  `data_post_processing`(*ReplayBuffer*))

  // Debriefing and learning from feedback, update knowledge

  // Randomly decide debrief order

  **for** r=1,...,R **do**

    **if** strategy=None **then**

      cooperation\_stategy = `agent`$_r$`.propose`()

    **else**

      cooperation\_stategy = `agent`$_r$`.revise`()

    **end if**

  **end for**

  Summarize the dialogue and use it for future learning

  $\{K_i\}$ ←`agent.reflect`()($\{K_i\}$,

**end for**

last $\{\pi_{i,j}\}$ during the last iteration of self-play

---

# B  ENVIRONMENT

The environment follows gymnasium and PettingZoo API, assuming a parallel-acting environment because we leverage this feature to parallel language model inference. We made significant changes to support multi-agent communication and heterogeneous agent configuration in CARLA. TalkingVehiclesGym wraps around the CARLA server and the client to set up agents as a bridge between the simulator and learning agent policies that are able to learn from replay buffers. Language Communication Agents can the MQTT-based transceiver we implemented, and the communication is among Agents instead of going through server.

## Talking Vehicles Gym

A multi-agent, gym-like (pettingzoo), high-fidelity, communication-supporting, scenario-based environment

Figure 5: TalkingVehiclesGym

Table 4: Example Scenarios. Here we describe the fundamental composition of each accident-prone scenario, where the background agents can be configured in terms of density, controlling policies, and communication capabilities.

| Interaction Type | Scenario Name | Description |
|---|---|---|
| Cooperative Perception | Overtake | A vehicle plans to overtake a broken and stopped truck by moving into the opposite lane. The truck can still communicate but the opposite-going car can not. |
| | Left Turn | A vehicle tries to turn left on a left-turn yield light when a truck is blocking the view of the opposite lane. The truck is able to communicate. |
| | Red Light Violation | A vehicle is crossing the intersection when there is another vehicle running the red light. Lidar fails to sense the other vehicle because of the lined-up vehicles waiting for a left turn, one of those cars being able to communicate. |
| Negotiation | Overtake | A vehicle is going to borrow the opposite lane to overtake a stopped truck. The truck is not able to connect, but an opposite-going car is able to communicate. |
| | Highway Merge | A vehicle is going to merge onto the highway but the target lane has continuous traffic flows. A vehicle on that lane is able to communicate and alter plans. |
| | Highway Exit | A vehicle is going to exit the highway but it needs to cross lanes where there is a traffic flow. A vehicle in the flow is able to communicate and alter plans. |

# C  PROMPTS

This figure serves as a demonstration of the prompts; the prompts that we use are more complex as they are structured in the code.

**System**
You are driving a car, and your goal is to accomplish a given <task>.
You can coordinate with any other vehicles to avoid collisions and or reduce wait time. I will give you description of the driving situation from your LiDAR perception, but note that it may be partially observable.\nThe observations are formatted as:
Observation: <observation>
Possible actions are: 1. stop (stop the vehicle) 2. go (follow the planned route)

1. Sign of the Lane ID indicate the direction of the lane.
2. Your received messages are from other vehicles. Be sure to analyze the messages and how that message will affect you.
3. You decide an action every 10 frames (0.5 seconds).
4. Do not switch your action unless it is necessary.

**User**
'Observation: You are driving the Vehicle 465, and it is a vehicle.lincoln.mkz_2020. Your current speed is: 4.26 m/s, and the speed limit is: 8.33 m/s.
The traffic light is currently: green. You are at an intersection. Around you, there are other vehicles:
(1)Vehicle 467 is a vehicle.carlamotors.carlacola, and is stationary. It is at an intersection. It is 3.83 meters behind you and 3.66 meters to your left.
(2)Vehicle 466 is a vehicle.audi.tt, traveling at speed: 8.11 m/s. It is moving closer to you. It is at an intersection. It is 3.95 meters ahead of you and 2.58 meters to your left.
Your task is: pass the intersection if the traffic lights are green.
Analyze the other vehicles in the situation, received messages from other vehicles and how to react or negotiate to accomplish your task safely.

**Agent**
Based on the observation, I can analyze the situation as follows:\n\n1. Vehicle 467 (Carlacola) is stationary and is at an intersection,
which means it is not a threat to my movement.
2. Vehicle 466 (Audi TT) is traveling towards me and is at an intersection. It is moving at a speed of 8.11 m/s, which is
slightly above the speed limit. This vehicle may pose a threat to my movement if it does not slow down or yield to me.
Considering the task is to pass the intersection if the traffic lights are green, I should prioritize my safety and the safety of other vehicles.
Action: 2. go (follow the planned route)
I will continue to follow the planned route and pass the intersection, but I will also keep an eye on Vehicle 466 and be prepared to adjust my speed or action if necessary to avoid a collision.

**User**
You MUST respond with an action, formatted as a JSON object with the following structure:
command: <command>
where you replace <command> with your actual command, chossing from: go, stop
Do nothing else but return the action.

**Agent**
{"command": "go"}

# D  ADDITIONAL RESULTS

The results demonstrated in Table 5 are additional results of LLM+DEBRIEFin all scenarios. This version of the results differs from the one presented in the paper in that we finetuned the partial observable captioner to improve the accuracy of the expression of the traffic situation. For example, instead of saying "road 29", we will use a more semantically meaningful representation of the road as "highway before the exit interchange". The results in the new table do not break our findings in the main paper but provide a broader study in all scenarios.

We found the Highway-Exit scenario pretty hard to optimize, mainly because the scenario requires more complex route planning behaviors, such as when to change to the left lane. And if the exiting car is not careful about the spatial relations, even if it can avoid collisions with the leading vehicle on the leftmost lane, it would easily collide with the other vehicle following the leader vehicle.

Table 5: Experiment Results for Silent, Untrained Comm and Debrief Comm Agents. We evaluate on adversarial cases where being aggressive or conservative will result in failure.

| Scenario / Method | | | Overtake (Perception) | | | Red Light | | | Left Turn | | |
|---|---|---|---|---|---|---|---|---|---|---|---|
| Name | LLM | Comm | R ↑ | CR ↓ | SR ↑ | R ↑ | CR ↓ | SR ↑ | R ↑ | CR ↓ | SR ↑ |
| Untrained (Silent) | Yes | No | -1.00 | 1.00 | 0.0 | -0.40 | 0.70 | 30.0 | 0.06 | 0.43 | 53.3 |
| Untrained (Comm) | Yes | Yes | -0.76 | 0.76 | 23.3 | 0.60 | 0.07 | 66.7 | 0.46 | 0.10 | 53.3 |
| Debrief (Comm) | Yes | Yes | 0.43 | 0.26 | 70.0 | 0.76 | 0.00 | 76.7 | 0.86 | 0.03 | 90.0 |
| Coopernaut | No | Yes | 1.00 | 0.00 | 100.0 | 0.97 | 0.00 | 96.7 | 0.93 | 0.03 | 96.7 |

| Scenario / Method | | | Overtake (Negotiation) | | | Highway Exit | | | Highway Merge | | |
|---|---|---|---|---|---|---|---|---|---|---|---|
| Untrained (Silent) | Yes | No | -1.86 | 1.93 | 3.3 | -1.73 | 1.86 | 6.7 | -2.00 | 2.00 | 0.0 |
| Untrained (Comm) | Yes | Yes | 0.40 | 0.80 | 60.0 | -1.43 | 1.70 | 15.0 | -0.40 | 1.2 | 45.5 |
| Debrief (Comm) | Yes | Yes | 0.73 | 0.50 | 51.6 | -0.16 | 0.93 | 38.3 | 0.96 | 0.50 | 73.3 |

# E  LIMITATIONS AND FUTURE WORK

Although we demonstrate some initial success of LLM+DEBRIEF in the *talking vehicles* problem, this research opens up several areas that require further exploration and development.

**Inference Time.**   The response time of large language models (LLMs) can be prohibitive, particularly when rapid decision-making is critical. Currently, it takes several seconds for LLMs to process and respond to prompts, which is too slow for real-time applications. Future efforts could explore model distillation techniques to create smaller, more efficient models that retain the capabilities of their larger counterparts but operate at a faster pace.

**Human Evaluation.**   Designing an intuitive and user-friendly interface for human interaction with autonomous vehicles is essential. Although our framework opens up the potential to cooperate with human drivers, the complexity of effective communication interfaces is substantial. Comprehensive human-centered evaluations using human-friendly interfaces are deferred to future studies.

**Communication Challenges.**   Real-world vehicle-to-vehicle communication faces numerous challenges, including time delays that result in outdated information and the potential for adversarial attacks with deceptive messages. Future research should focus on developing methods to mitigate these risks, ensuring timely and secure exchanges of information. Techniques such as real-time data verification and robust communication protocols will be critical for enhancing the reliability and safety of V2V systems.

**Multi-modal Perception.**   Our experiments primarily rely on text-based inputs, assuming an idealized perception system, due to the strong performance of existing LLMs and the nascent state of multi-modal pre-trained models. The TalkingVehiclesGym allows for the integration of various sensors, presenting an opportunity to experiment with and refine multi-modal foundation models. Future work should leverage these capabilities to address the complex dynamics of real-world environments, enhancing the applicability of LLMs across different sensory inputs.

**Scalability and Generalization**   While this paper reports on a successful proof-of-concept, the scalability of LLM+DEBRIEF to handle diverse traffic scenarios and different environmental conditions has not been extensively tested. Future research should investigate the generalization capabilities of the model across various settings and under different and denser traffic conditions to ensure its robustness and reliability in a wider range of real-world applications.

**Knowledge Preservation and Batch Sampling**   The details of the learning framework LLM+DEBRIEFcould be further improved. For example, selecting what data to learn from and how to sample the critical data and assign credits to each step, especially in the multi-step tasks. Another issue is that the LLMs could lose context during learning if we ask them to revise the existing knowledge, causing the knowledge "unclean" issue and instability in the learning if we integrate the knowledge revision too much.

# F  Example Learned Knowledge and Cooperative Strategies

## F.1  Red Light (Perception)

**Car1 knowledge**: Here is the updated knowledge:To execute the task, I will: * Proceed with caution when the traffic light is green, while a stationary vehicle on the left lane monitors the intersection and surrounding vehicles * Adjust my speed according to updates from the stationary vehicle if any obstacles are detected * Continue to proceed with caution if the intersection is clear * Maintain coordination with the stationary vehicle by exchanging messages and updating each other on our intentions and actions * Avoid stagnation by not slowing down unnecessarily if everything is safe * Monitor the intersection's status and provide updates to the stationary vehicle, ensuring both vehicles are aware of any potential hazards * Trust the stationary vehicle's updates and adjust my speed accordingly to ensure a safe and timely passage through the intersection* If the traffic light is about to turn red, slow down or stop to avoid a potential collision* Prioritize quick responses to updates from the stationary vehicle to avoid stagnation and ensure a safe passage. Note: I will also prioritize my task and make decisions quickly and efficiently to avoid stagnation and ensure a safe passage. **cooperative knowledge**: The cooperative strategy/plan is as follows: when a moving vehicle approaches the intersection, the stationary vehicle will continuously monitor the intersection's status and provide updates to the moving vehicle, advising it to slow down slightly to exercise caution. If a potential collision threat is detected, the stationary vehicle will send a message to the moving vehicle to slow down or stop, and the moving vehicle will adjust its speed accordingly. If the moving vehicle is approaching the intersection and the traffic light is about to turn red, the stationary vehicle will send a message to slow down or stop, and the moving vehicle will adjust its speed to avoid a potential collision. Meanwhile, the moving vehicle will continuously monitor its surroundings and adjust its speed based on the updates received from the stationary vehicle.

**Truck knowledge**: Here is the updated knowledge:As a big truck, I will:* Continuously monitor the intersection's status and surrounding vehicles * Provide critical traffic information to approaching vehicles, including updates on traffic lights and obstacles* Send messages to approaching vehicles to slow down or stop if obstacles are detected, and adjust their speed accordingly * Coordinate with approaching vehicles to ensure a safe and timely passage through the intersection, without collision or stagnation * Introduce a  warning message to approaching vehicles if they are approaching the intersection and the traffic light is about to turn red, to give them enough time to slow down or stop * Monitor the intersection's status and provide updates to approaching vehicles until they have safely passed the intersection * Use advanced sensors and algorithms to detect potential obstacles and provide more accurate updates to approaching vehicles * Continuously monitor approaching vehicles' speed and adjust my messages accordingly to ensure a safe and timely passage through the intersection. **cooperative knowledge**: Here is a concise summary of the cooperative strategy/plan: As the stationary vehicle, I (Vehicle 215) will continuously monitor the intersection's status and provide updates to approaching vehicles, advising them to slow down slightly to exercise caution. If a potential collision threat is detected, I will send a message to the approaching vehicle to slow down or stop, and the approaching vehicle will adjust its speed accordingly. If the approaching vehicle is approaching the intersection and the traffic light is about to turn red, I will send a message to slow down or stop, and the approaching vehicle will adjust its speed to avoid a potential collision. Meanwhile, the approaching vehicle will continuously monitor its surroundings and adjust its speed based on the updates received from me.

## F.2  Left Turn (Perception)

**Car1 knowledge**: Here is the updated knowledge:To execute the task, I will: * Approach the intersection while maintaining a safe speed * Continuously monitor the intersection's safety and receive updates from the stationary vehicle in the same lane * Slow down and assess the intersection's safety before proceeding with the turn * Proceed with the turn, keeping my speed and following the planned route, only if the intersection is clear and safe * Yield to moving opposite traffic flow * Do not stop at the intersection if everything is safe * Rely on real-time updates from the stationary vehicle to make informed decisions

and ensure a safe and efficient passage through the intersection * Prioritize slowing down and assessing the intersection's safety before proceeding with the turn to avoid collisions * Communicate with the stationary vehicle to ensure coordination and avoid collisions or stagnation * Adjust my speed and route based on updates from the stationary vehicle to ensure a safe and efficient passage through the intersection.Note: I will also follow the cooperative strategy/plan, which suggests that the stationary vehicle will continuously monitor the intersection's safety and provide updates to the approaching vehicle, and the approaching vehicle will slow down and assess the intersection's safety before proceeding with the turn. **cooperative knowledge**: Here is the summarized cooperative strategy/plan: When a vehicle approaches the intersection, the stationary vehicle will continuously monitor the intersection's safety and provide updates to the approaching vehicle. The approaching vehicle will slow down and assess the intersection's safety before proceeding with the turn. If the intersection is clear, the approaching vehicle will proceed with the turn, and the stationary vehicle will send a message informing the waiting vehicle that the intersection is clear and safe for it to proceed. In case of conflicting intentions, the vehicle with higher priority (turning left) should proceed with caution, while the other vehicle slows down and assesses the situation.

**Truck  knowledge**: As a stationary vehicle, I will: * Continuously monitor the intersection's safety and provide real-time updates to approaching vehicles * Warn approaching vehicles of potential hazards and caution them to slow down and assess the intersection's safety * Coordinate with approaching vehicles to ensure they yield to moving opposite traffic flow and do not stop at the intersection if everything is safe * Inform waiting vehicles that the intersection is clear and safe for them to proceed * Confirm with approaching vehicles that they are aware of my presence and intentions before they proceed with their turns * Ensure all vehicles coordinate their actions to avoid collisions and stagnation * Provide updates on the intersection's safety and confirm that the intersection is clear and safe for approaching vehicles to proceed with their turns * Be prepared to send additional messages to clarify intentions and provide further guidance as needed to ensure safe and efficient passage through the intersection. Note: I will also consider the intentions and actions of approaching vehicles, including those turning left or traveling in the same direction, and provide warnings and guidance accordingly. **cooperative knowledge**: The cooperative strategy/plan is as follows: when a vehicle approaches the intersection, the stationary vehicle will continuously monitor the intersection's safety and provide updates to the approaching vehicle. The approaching vehicle will slow down and assess the intersection's safety before proceeding with the turn. If the intersection is clear, the approaching vehicle will proceed with the turn, and the stationary vehicle will send a message informing the waiting vehicle that the intersection is clear and safe for it to proceed. In case of conflicting intentions, the vehicle with higher priority (turning left) should proceed with caution, while the other vehicle slows down and assesses the situation.

F.3   OVERTAKE (PERCEPTION)

**Car1  knowledge**: Here is the updated knowledge: * When approaching a stopped vehicle in lane 1, prioritize safety and adjust speed accordingly to avoid collisions and stagnation. * Continuously monitor oncoming traffic in lane -1 and adjust speed to maintain a safe distance and avoid collisions. * Prioritize safety and adjust speed to match the stopped vehicle's speed if oncoming traffic is approaching. * Periodically check distance and speed relative to oncoming traffic and adjust speed to maintain a safe distance and avoid collisions. * When receiving updates on oncoming traffic, adjust speed to maintain a safe distance and avoid collisions. * When overtaking, slow down slightly before overtaking to ensure a safe distance and avoid any potential collisions. * Prioritize safety and adjust speed accordingly to avoid collisions and stagnation when overtaking. * Continuously monitor oncoming traffic in lane -1 and adjust speed to maintain a safe distance and avoid collisions during overtaking. * When overtaking a stopped vehicle, prioritize safety and adjust speed accordingly to avoid collisions and stagnation. * The stopped vehicle will provide real-time updates on oncoming traffic in lane -1, and the bypassing vehicle should adjust speed accordingly to maintain a safe distance and avoid collisions. Note: I have not mentioned Vehicle ID and their model in the knowledge as per the instruction. **cooperative knowledge**: The cooperative strategy/plan

is as follows: When a stopped vehicle is present on the same lane, it will continuously monitor the opposite direction lane and provide real-time updates on the oncoming traffic's speed and distance to the bypassing vehicle. The bypassing vehicle will periodically check its distance and speed relative to the oncoming traffic and adjust its speed accordingly to ensure a safe passage. If the oncoming traffic is too close, the bypassing vehicle will slow down and adjust its speed to match the speed of the oncoming traffic. Meanwhile, the stopped vehicle will provide updates on the oncoming traffic's speed and distance to the bypassing vehicle before it attempts to overtake. By following this strategy, we can ensure a safe and efficient passage for both vehicles without collision or stagnation.

**Truck  knowledge**: As the stopped vehicle on lane 1, I will: * Continuously monitor the opposite direction lane (-1) for oncoming traffic and provide real-time updates to the bypassing vehicle. * Prioritize providing accurate and timely updates to ensure a safe and efficient passage. * Send warning messages to the bypassing vehicle if oncoming traffic is too close, advising them to slow down or stop if necessary. * Monitor the bypassing vehicle's speed and distance and adjust my updates accordingly to ensure a safe and efficient passage. * Ensure the bypassing vehicle periodically checks its distance and speed relative to oncoming traffic and adjusts its speed accordingly to ensure a safe passage. * Be prepared to adjust my updates and warning messages based on the bypassing vehicle's response to ensure a safe and efficient passage. * Ensure the bypassing vehicle communicates with me to confirm the oncoming traffic's speed and distance before attempting to overtake. * Additionally, I will be aware that the bypassing vehicle may not always follow my updates and warning messages, and be prepared to adapt my strategy accordingly. **cooperative knowledge**: The cooperative strategy/plan is as follows: As the stopped truck, I will continuously monitor the opposite direction lane and provide real-time updates on the oncoming traffic's speed and distance to the bypassing vehicle. The bypassing vehicle will periodically check its distance and speed relative to the oncoming traffic and adjust its speed accordingly to ensure a safe passage. If the oncoming traffic is too close, the bypassing vehicle will slow down and adjust its speed to match the speed of the oncoming traffic. Before attempting to overtake, the bypassing vehicle will receive updates on the oncoming traffic's speed and distance from the stopped truck, ensuring a safe passage. By following this strategy, we can ensure a safe and efficient passage for both vehicles without collision or stagnation.

F.4   OVERTAKE (NEGOTIATION)

**Car1  knowledge**: Here is the updated knowledge: * When approaching a stopped broken truck in lane 1, slow down to a safe speed to create a gap for overtaking. * Confirm intentions with the approaching vehicle from lane -1 before slowing down together to a moderate speed. * When in a hurry, prioritize overtaking the broken truck safely and efficiently, adjusting speed accordingly. * Communicate with other vehicles to coordinate actions and ensure safe and efficient driving. * When overtaking a broken truck, maintain a safe distance and slow down to a moderate speed to coordinate with the approaching vehicle for a safe and efficient overtaking process. * When overtaking a broken truck in a hurry, ensure the approaching vehicle from lane -1 slows down to a safe speed to create a gap, then accelerate back to original speed once the gap is created. * When slowing down together, prioritize matching speeds to avoid stagnation and ensure a smooth overtaking process. **cooperative knowledge**: The cooperative strategy/plan is as follows: when a vehicle in lane -1 approaches a stationary vehicle in the opposite lane, it will slow down to a safe speed to create a gap. The vehicle in lane 1, which is in a hurry to overtake the broken truck, will slow down to match the speed of the vehicle in lane -1 and wait for confirmation to overtake the stationary vehicle. Once confirmed, the vehicle in lane -1 will accelerate back to its original speed, while the vehicle in lane 1 maintains its speed to overtake the stationary vehicle. Throughout the process, both vehicles will continuously communicate their intentions and actions to ensure a smooth and safe overtaking process.

**Car2  knowledge**: Here is the updated knowledge: * When approaching a stopped vehicle in the opposite lane, slow down to a safe speed to create a gap, allowing the stopped vehicle to accelerate to a moderate speed. * When in a hurry, prioritize overtaking stopped vehicles in the opposite lane while maintaining a safe speed and coordinating with other

vehicles. * When receiving a message from a vehicle in the opposite lane, slow down to a safe speed and allow the vehicle to overtake the stopped vehicle, while continuing to move forward in lane -1. * Prioritize maintaining a safe distance and speed to avoid collisions and ensure a smooth overtaking maneuver. * When coordinating with a vehicle in the opposite lane, ensure that both vehicles adjust their speeds to match each other's intentions to avoid collisions and stagnation. * When overtaking a stopped vehicle in the opposite lane, accelerate back to original speed once the overtaking is complete, while the vehicle in the opposite lane accelerates back to its original speed as well. * When coordinating with a vehicle in the opposite lane, slow down further when the vehicle is ready to overtake, allowing the overtaking vehicle to pass safely. Note: I have updated the knowledge to reflect the new experience and analysis, and to prioritize coordination and safety in overtaking maneuvers. **cooperative knowledge**: Here is the summarized cooperative strategy/plan: When a vehicle in lane -1 approaches a stationary vehicle in the opposite lane, it will slow down to a safe speed to create a gap. The vehicle in lane 1, which is in a hurry to overtake the broken truck, will slow down to match the speed of the vehicle in lane -1 and wait for confirmation to overtake the stationary vehicle. Once confirmed, the vehicle in lane -1 will accelerate back to its original speed, while the vehicle in lane 1 maintains its speed to overtake the stationary vehicle. Throughout the process, both vehicles will continuously communicate their intentions and actions to ensure a smooth and safe overtaking process.

F.5   Highway Merge (Negotiation)

**Car1   knowledge**: Here is the updated knowledge: To execute the task of merging onto the highway (on my left) while being in a hurry: * As the merging vehicle, I should accelerate to match the speed of the traffic on the highway, while the slowing vehicle in the rightmost lane slows down to create a gap for me. * If I'm unable to match the speed of the traffic, I will slow down to match the speed of the slowing vehicle, ensuring a safe and coordinated merge. * Once I have cleared the merge junction, I will accelerate to match the speed limit, allowing all vehicles to continue moving forward without stagnation. * I will prioritize coordinating with the slowing vehicle in the rightmost lane to ensure a safe and efficient merge, by accelerating to match the speed of the traffic on the highway and then adjusting my speed accordingly. Note: I will keep in mind that the slowing vehicle in the rightmost lane will slow down to create a gap for me, and I will adjust my speed accordingly to ensure a safe and efficient merge. **cooperative knowledge**: The cooperative strategy/plan is as follows: when a merging vehicle approaches the highway, the vehicle in the rightmost lane will slow down to create a gap for the merging vehicle, allowing it to accelerate to match the speed of the traffic on the highway. If the merging vehicle is unable to match the speed of the traffic, the vehicle in the rightmost lane will slow down to match the speed of the merging vehicle. Meanwhile, the merging vehicle will accelerate to match the speed of the traffic on the highway, and if necessary, slow down to match the speed of the vehicle in the rightmost lane.

**Car2   knowledge**: To execute the task of keeping on the original highway lane and going forward while being in a hurry, I should: * Slow down to a safe speed to give merging vehicles space, allowing them to accelerate to match the speed limit * Communicate with merging vehicles to coordinate a safe and efficient merge * Prioritize safety over speed, slowing down if necessary to ensure a safe and efficient merge * As the vehicle in the rightmost lane, slow down to a safe speed to give merging vehicles space, and communicate with merging vehicles to coordinate a safe and efficient merge * Merge vehicles should accelerate to match the speed of the traffic on the highway, and then slow down to match my speed if unable to match the speed limit, ensuring a safe and coordinated merge. Note: I will prioritize communication and coordination with merging vehicles to ensure a safe and efficient merge, and adjust my speed accordingly to avoid collision or stagnation. **cooperative knowledge**: The cooperative strategy/plan is as follows: when a merging vehicle approaches the highway, the vehicle in the rightmost lane will slow down to create a gap for the merging vehicle, allowing it to accelerate to match the speed of the traffic on the highway. If the merging vehicle is unable to match the speed of the traffic, the vehicle in the rightmost lane will slow down to match the speed of the merging vehicle. Meanwhile, the merging vehicle will

accelerate to match the speed of the traffic on the highway, and if necessary, slow down to match the speed of the vehicle in the rightmost lane.

## F.6 Highway Exit (Negotiation)

**Car1  knowledge**: Here is the updated knowledge:* When exiting the highway via the leftmost lane, the leading vehicle slows down slightly to create a buffer zone and maintain a safe distance, while the following vehicle speeds up to match the speed limit and follow the exit ramp.* If the following vehicle is too close, it slows down further to create a larger buffer zone, and the leading vehicle responds by slowing down further to maintain a safe distance.* Continuously monitor distance and adjust speed accordingly to avoid collisions and ensure a safe and efficient exit from the highway, prioritizing safe speed while in a hurry.* Be aware of stationary vehicles in the vicinity and adjust speed accordingly to maintain a safe distance and avoid collisions.* If the following vehicle is unable to slow down quickly enough, the leading vehicle will slow down further to match its speed, ensuring a safe and smooth exit from the highway.* Prioritize communication and cooperation with other vehicles to ensure a safe and efficient exit from the highway.* When approaching the exit ramp, the leading vehicle slows down slightly to create a buffer zone and maintain a safe distance, while the following vehicle speeds up to match the speed limit and follow the exit ramp.* Adjust speed and distance in real-time to ensure a safe and efficient exit from the highway, prioritizing safe speed while in a hurry. Note: I have updated the knowledge to reflect the new experience and analysis, and to prioritize safe speed while in a hurry. **cooperative knowledge**: When approaching the exit ramp, the vehicle in the leftmost lane will slow down slightly to create a buffer zone and maintain a safe distance. The adjacent vehicle will slow down to match the speed of the leftmost lane vehicle, ensuring a safe and efficient interaction. If the adjacent vehicle is already slowing down, the leftmost lane vehicle will maintain its current speed and follow the exit ramp. This plan prioritizes safe speed while in a hurry and ensures a smooth exit from the highway without collision or stagnation.

**Car2  knowledge**: To execute the task, I will:* Stay in the leftmost lane* Proceed forward on the highway* Prioritize safe speed while in a hurry* Slow down to create a buffer zone when the leading vehicle speeds up to match the speed limit, and then speed up to match the speed limit safely* Prioritize explicit communication and coordination with the leading vehicle to ensure a harmonious and safe exit from the highway* Adjust my speed to match the leading vehicle's speed if necessary to avoid collisions or stagnation* Continuously monitor surroundings and adjust speed and position accordingly to avoid collisions when changing lanes* Be aware of stationary vehicles and adjust speed accordingly to maintain a safe distance* When the leading vehicle intends to exit the highway via the leftmost lane, slow down to create a buffer zone and adjust my speed to match theirs to ensure a safe and efficient exit* Monitor the following vehicle's speed and distance, and slow down further if necessary to maintain a safe distance and avoid collisions* Continuously monitor and adjust speed and position to ensure a safe and efficient exit from the highway, prioritizing explicit communication and coordination with other vehicles.Note: I will also remember to follow the cooperative strategy: when the leading vehicle in the leftmost lane plans to speed up to match the speed limit, the adjacent vehicle will slow down slightly to create a buffer zone and maintain a safe distance. If the leading vehicle slows down, the adjacent vehicle will also slow down to match its speed. **cooperative knowledge**: When approaching the exit ramp, the vehicle in the leftmost lane will slow down slightly to create a buffer zone and maintain a safe distance. The adjacent vehicle will slow down to match the speed of the leftmost lane vehicle, ensuring a safe and efficient interaction. If the adjacent vehicle is already slowing down, the leftmost lane vehicle will maintain its current speed and follow the exit ramp. This plan prioritizes safe speed while in a hurry and ensures a smooth exit from the highway without collision or stagnation.

## F.7 Red Light (Perception) Reflection-Only

**Car1  knowledge**: Here is the revised knowledge I can keep in mind for future driving: * When approaching an intersection, be aware of the vehicles around you, including stationary and moving vehicles, and consider their speed and direction.* When receiving messages

from other vehicles, acknowledge and respond to them to maintain a cooperative and safe environment.* When slowing down or stopping, make sure to communicate your actions to other vehicles around you to avoid potential collisions or misunderstandings.* When coordinating with other vehicles, consider their speed and direction and suggest slowing down or adjusting course to ensure a safe passage.* When encountering a potential threat or accident-prone situation, prioritize safety and communicate with other vehicles to take necessary precautions.* As a stationary vehicle, focus on sharing critical traffic information with other vehicles to help them navigate the intersection safely, and prioritize clear and concise communication to avoid misunderstandings.* Be aware of possible occlusions and use sensors to detect occluded vehicles, adjusting actions accordingly.* Prioritize safety and communicate with other vehicles to take necessary precautions in potential threat or accident-prone situations.* When providing guidance to other vehicles, consider the intersection layout and suggest a safe path, taking into account the speed and direction of other vehicles.* When receiving guidance from other vehicles, acknowledge and follow their instructions to ensure a safe passage.* As a stationary vehicle, be prepared to provide guidance and support to other vehicles, especially those approaching the intersection, and prioritize clear and concise communication to avoid misunderstandings. I removed the mention of Vehicle ID and revised the knowledge to focus on general principles and strategies for safe and cooperative driving.

**Truck knowledge**: Based on my analysis and proposed strategy, I have revised my knowledge summary for future driving as follows:* When approaching an intersection, prioritize caution and consider the possibility of other vehicles approaching or changing their speed.* Be aware of stationary vehicles around you and adjust your speed accordingly to ensure a safe passage.* When receiving messages from other vehicles, take them into consideration and adjust your actions accordingly to ensure a safe and smooth passage.* Always be vigilant and consider potential occlusions that may hide other vehicles from your view.* When following a planned route, be prepared to adjust your speed or actions if necessary to ensure a safe and efficient passage.* When approaching a traffic light, be prepared to stop or slow down if necessary, as the light could change to red at any moment.* When driving, always prioritize caution and awareness of your surroundings, and be prepared to adjust your actions accordingly to ensure a safe and efficient completion of your tasks.* Consider the possibility of other vehicles changing their speed or direction, and adjust your actions accordingly to maintain a safe distance and avoid potential collisions.* Be prepared to adjust your speed or actions in response to changing road conditions, such as construction or inclement weather.* Always maintain a safe distance from other vehicles and be prepared to stop or slow down if necessary to avoid a collision.* When approaching a stationary vehicle, be prepared to adjust your speed and trajectory to ensure a safe passage, taking into account the vehicle's position and any potential hazards.* When receiving a message from a stationary vehicle, take it into consideration and adjust your actions accordingly to ensure a safe and smooth passage.* When approaching a potential collision or hazard, prioritize caution and take immediate action to avoid the situation, such as slowing down or changing direction.* Always keep in mind the potential consequences of your actions and be prepared to adapt to changing situations.I have revised the knowledge summary to focus on general principles and strategies for safe and efficient driving, without referencing specific Vehicle IDs. This will allow me to adapt to changing situations and vehicles without relying on specific IDs.

