# OpenReview forum: "Talking Vehicles: Cooperative Driving via Natural Language"
_ICLR.cc/2025/Conference — Submitted to ICLR 2025_

### Official Review · Reviewer_tgAQ · 2024-10-24

**Soundness:** 2
**Presentation:** 2
**Contribution:** 2
**Rating:** 3
**Confidence:** 3

**Summary:**

The paper proposed using natural language as the means of communication in traffic scenarios with V2V communication. In particular, the authors found that LLM + debrief significantly improves the performance of the cooperative planning pipeline. The proposed method is evaluated in a gym-like simulation environment equipped with LLM and Carla as the vehicle simulator and showed some signs of life. Comparing to the baseline, using language feedback leads to much smaller communication messages yet the performance seems worse than the baseline.

**Strengths:**

1. The idea of using natural language as feedback is interesting. Although it is probably considered by many people, the authors managed to build a simulation environment that enables communication with natural language.

**Weaknesses:**

1. The idea of isolating a finite set of agents called the "focal group" is not really practical in real world as the communication graph can extend to a huge size, although I understand that this is probably the common setup for V2V communication scenarios.
2. I find it very confusing as to which part of the simulation environment is automated and which part is hand-crafted/hard-coded. I think this is critical to assessing the practicality of the proposed method and a more detailed breakdown would be nice.
3. The concept of a partially observed general-sum game isn't really relevant to the proposed method. I understand that it is a nice way to describe the problem, but I don't see any game theory tools used in the actual solution.
4. The performance of the LLM agent is worse than the baseline. Although the messages were smaller, I wonder whether the benefit of smaller messages is overshadowed by the significant increase in computation. More analysis on this is needed.

**Questions:**

1. Please clearly state which part of the simulator is fully automated and which part is hand-crafted/hard-coded.
2. What is the role of game theory in the proposed method?

---

> ### Author Response · Authors · 2024-11-22
> **Official Comments by Authors [1/2]**
>
> Dear Reviewer tgAQ
>
> Thank you for your insightful comments and questions. We sincerely appreciate the opportunity to address your insightful comments and clarify aspects of our work. Below, we provide detailed responses to each of your points:
>
> ---
>
> **1. Game Theory Notation, Focal Group, and Communication Graph**
> While our work is not rooted in game theory, we describe it as a partially observable stochastic game for notational clarity and consistency with previous literature [1-3]. Additionally, we formally define the collaboration property of the focal group, as introduced in [1]. By definition, the focal group operates collaboratively against a diverse set of potentially unseen agents. In our current formulation, other agents are assumed to be truthful and willing to collaborate if communication is possible, though this assumption could be relaxed in future studies. The concept of a "focal group" is not specific to Vehicle-to-Vehicle (V2V) communication but is introduced here to facilitate future research on the generalizability of focal group policies. Communication in our framework relies on a broadcasting mechanism, with agents specifying recipients in the message content. This means communication is inherently influenced by the proximity of agents, and agents (communication nodes) outside the focal group can also be part of the communication graph. This design supports flexibility and extensibility in studying multi-agent communication and coordination.
>
> **2. Which part of the simulation is hand-crafted and hard-coded**
> We would appreciate it if the reviewer could explain this question further if the following response does not answer your question well. We aim to make the framework as general as possible and use a [hydra configuration system](https://hydra.cc/), so most of our modules are **“automated”** or **“configurable”**. The simulator generally wraps the core scenario in CARLA as a [multi-agent gym environment](https://anonymous.4open.science/r/talking-vehicles/code/envs/multiagent_env.py). We basically contributed all the code in the submitted code repository, which depends on the CARLA Python API and utility functions from https://github.com/carla-simulator/scenario_runner that are not included in the code folder.
>
> **“Hard-coded Part in the Simulator and Framework”**
> We hypothesize that the reviewer refers to “hard-coded” components of some modules that could have been configured differently.
> - In our simulation framework, we dedicated significant effort to developing a **partial observation captioner** [[partially_observable_captioner.py]](https://anonymous.4open.science/r/talking-vehicles/code/utils/partial_observable_captioner.py), which translates simulator information into text descriptions *as accurately as possible*. This modularized component provides a detailed representation of the environment, including vehicle states, lane markings, and descriptions of visible surrounding vehicles by directly accessing simulator data. The visibility of a vehicle is computed by our geometry calculator  [[shapley_geometry.py]](https://anonymous.4open.science/r/talking-vehicles/code/utils/shapely_geometry.py). By using the captioner, users can bypass the need to process raw sensor data, allowing them to focus on the core research question (as discussed in **General Response Q1**).
> - We also modularized the **chain-of-thought prompting** within the framework, equipping it with common knowledge, rules, and questions to facilitate reasoning in the LLM-based policy [[llm_policy.py](https://anonymous.4open.science/r/talking-vehicles/code/policies/llm_policy.py), [prompts.py](https://anonymous.4open.science/r/talking-vehicles/code/policies/utils/prompts.py)]. This design enhances the decision-making process of the LLM policy by embedding structured guidance. Future work could study the general prompting scheme or LLM policy.
> - In the scenarios studied, agents are configured with individual target locations and tasks but are configurable for future studies (such as “Exit highway via the leftmost lane and follow the exit ramp, prioritizing safe speed while in a hurry.” [[example_config]](https://anonymous.4open.science/r/talking-vehicles/code/envs/scenarios/multiagent_scenarios/highway_exit/highway_exit__negotiation_comm_risky.yaml)). The preferences or tasks of agents are configured to create conflicts of interest, driving the exploration of coordination and communication strategies. We implement an atomic command follower [[atomic_command_follower.py]](https://anonymous.4open.science/r/talking-vehicles/code/agent/atomic_command_follower.py), which plans a reference route according to the target location and then translates high-level commands into actionable short-term path planning with the awareness of the global plan, such as "change to the left lane." in negotiation-highway-exit scenario alters where to change lanes and replan accordingly.

---

> ### Author Response · Authors · 2024-11-22
> **Official Comments by Authors**
>
> **3. Message Size Control vs Computation and Baselines**
> We selected two baselines[L401-415]: the previous LLM driving method, which uses retrieval-augmented methods [4], and the non-LLM, sensor-based method [5]. Our method is not directly comparable to [5], but we will discuss the analysis of the computation-message size tradeoff below:
> - Coopernaut [5] used LiDAR as sensor input and trained specific intermediate representation for communication along with the final control by imitating an expert whose representation is spatially meaningful. Such training methodology limits Coopernaut’s capability in 1. generalization (to agents that do not “speak” with the same encoder), 2. explainability, and 3. negotiation.
> - Our decision frequency is every 0.5 seconds with discrete high-level commands, while [5] decides every 0.05 seconds with fine-grained direct control supervision from an expert.  In contrast,  our method is completely based on self-play learning.
> - On the other hand, language messages are less expressive or accurate in terms of spatial information, and the agents are not as flexible as humans at **thinking from other agents’ perspectives**, as they are not trained for it. For example, in *Perception-Left-Turn*, it is hard for the truck to translate its perception of the environment to messages like “The is a car traveling southbound behind me on my right lane, which is the opposite traffic flow for you to take care of when you are turning left.”
> - We are aware of the limitations of context adaptation compared with gradient-based methods, which stably embed the experience into the model. We believe a combination of gradient-based method and self-play context adaption could further improve the methodology, and we leave it as a promising future work. (For reasons why we did not use a gradient-based method, please refer to the response to **Reviewer 9yf4, part 2** and General Response **Q3** for internalizing knowledge and saving computation of the models. )
>
> ---
>
> Thank you again for your valuable feedback. We hope our responses address your concerns and kindly request a revision of the ratings if our clarifications sufficiently address your questions. We are happy to provide further clarifications if needed.
>
> ### References
> [1] Agapiou, J. P., Vezhnevets, A. S., Duéñez-Guzmán, E. A., Matyas, J., Mao, Y., Sunehag, P., ... & Leibo, J. Z. (2022). Melting Pot 2.0. arXiv preprint arXiv:2211.13746.
>
> [2] Guo, S., Ren, Y., Mathewson, K., Kirby, S., Albrecht, S. V., & Smith, K. (2021).
> Expressivity of Emergent Languages is a Trade-off between Contextual Complexity and Unpredictability, International Conference on Learning Representations, 2022
>
> [3] Trivedi, R., Khan, A., Clifton, J., Hammond, L., Duéñez-Guzmán, E. A., Chakraborty, D., ... & Leibo, J. Z. Melting Pot Contest: Charting the Future of Generalized Cooperative Intelligence. In The Thirty-eight Conference on Neural Information Processing Systems Datasets and Benchmarks Track.
>
> [4] Wen, L., Fu, D., Li, X., Cai, X., Ma, T., Cai, P., ... & Qiao, Y. Dilu: A knowledge-driven approach to autonomous driving with large language models. The Twelfth International Conference on Learning Representations. 2024.
>
> [5] Cui, J., Qiu, H., Chen, D., Stone, P., & Zhu, Y. (2022). Coopernaut: End-to-end driving with cooperative perception for networked vehicles. In Proceedings of the IEEE/CVF Conference on Computer Vision and Pattern Recognition (pp. 17252-17262).

---

> ### Comment · Reviewer_tgAQ · 2024-11-25
>
> I appreciate the authors' response. My question regarding the simulator is clarified, it seems that the authors implemented a rule-based interface to convert the Carla output to language description. Regarding focal group and the game theory setup, I still have my concerns. The assumption that "operates collaboratively against a diverse set of potentially unseen agents" means that the agents' behaviors will depend on the definition of the focal group. To practically apply the proposed algorithm, one would need to partition all agents on the road to focal groups without overlap, which is not feasible in general IMO.
>
> I will raise my score to 4 thanks to the clarification re. the simulator.

---

> ### Author Response · Authors · 2024-12-02
>
> Thank you for your recognition of our explanation of the simulator and consideration for revising the score. We will further address your concern on the focal group and game set up.
>
> Your interpretation of a focal group is generally correct, a focal group, as promoted in [1, 3] in above references, is a group of agents who collaboratively work together within the group to achieve high total rewards / social welfare for the group, regardless of the background agents/policies outside the group. Just to confirm, your concern mainly lies in whether a vehicle could belongs to two focal groups, which requires it to act differently to cooperate.
>
> While we agree that partitioning vehicles into disjoint focal groups is impractical, our approach does not assume a centralized partitioning process for the following reasons:
> - The number of vehicles with V2V communication capabilities remains low, so collaboration is assumed to happen opportunistically. Consequently, the likelihood of a vehicle participating in multiple focal groups simultaneously is small.
> - In real-world scenarios, focal groups can form through quick collaboration commitments established via communication or pre-configured conventions by manufacturers. For instance, vehicles from the same company (e.g., Waymo cars and Cruise cars) may form focal groups without openly communicating with vehicles from other companies. If a vehicle faces the possibility of being part of multiple collaborations, it could employ a simple mechanism to decline additional commitments while engaged in an ongoing focal group
> - While the question of enabling overlapping focal groups is indeed interesting and challenging, it is beyond the scope of this paper. However, we provide an illustrative example with our scenario setups:
>
>     - **Scenario 1 (negotiation_overtake)**: In this setup, `car1` (the overtaking car) and `car2` (an opposing car) form a focal group to resolve a conflict caused by overtaking. Typically, `car2` stops to allow `car1` to overtake first ([[Video 1](https://anonymous.4open.science/r/talking-vehicles/video/negotiation_overtake_untrained_1.mp4)]); in other cases, the roles are reversed ([[Video 2](https://anonymous.4open.science/r/talking-vehicles/video/negotiation_overtake_untrained_2.mp4)]).
>
>      - **Scenario 2 (perception_overtake)**: Here, the setup involves `car1` as part of a focal group with a `truck` (a cooperative agent), while `car2` operates under a non-cooperative policy without communication. In this case, `car1` usually waits for `car2` to pass based on information from the truck before overtaking [[Video3](https://anonymous.4open.science/r/talking-vehicles/video/perception_overtake_debrief.mp4)].
>
>     - When both setups are evaluated together, car1 could possibly adopt a dominant strategy to wait for `car2` to pass first when it is fully informed about `car2`.
>
> We hope this clarification helps! The introduction of this concept might not be super useful in this work, but we introduce it here for the convenience of studying diverse background agent behaviors in the future. Thank you for engaging in this discussion and for your thoughtful comments!

---

### Official Review · Reviewer_9yf4 · 2024-10-26

**Soundness:** 1
**Presentation:** 2
**Contribution:** 2
**Rating:** 3
**Confidence:** 4

**Summary:**

This paper presents a suite of traffic coordination tasks for autonomous driving, formulated as situational communication within vehicle-to-vehicle settings. The objective is to use natural language to facilitate coordination, helping vehicles avoid imminent collisions and maintain efficient traffic flow. Authors introduced an LLM agent framework called LLM+Debrief, and developed a gym-like simulation environment featuring a range of accident-prone driving scenarios.

**Strengths:**

### New Angle on Vehicle-to-Vehicle Communication.

Unlike much of the existing work that emphasizes human-vehicle communication or latent (implicit) vehicle-vehicle signaling, this paper introduces a unique angle by explicitly utilizing natural language for vehicle-to-vehicle communication. This approach advances the field by demonstrating how explicit messaging can facilitate coordination and improve safety in multi-agent driving scenarios.

### Effective LLM+Debrief Framework
The proposed LLM+Debrief framework, trained over 30 episodes and evaluated on an additional 30. This episodic structure enhances the learning process and shows promise in learning teaming strategies among autonomous LLM agents.

### Contribution of a Gym-Like Simulation Environment
The development of a gym-like simulation environment focused on accident-prone driving scenarios represents a significant contribution to the field. If made publicly available, this environment could serve as a valuable resource for testing and advancing V2V communication protocols in autonomous driving research.

**Weaknesses:**

### Weakness 1: The motivation.
The motivation behind this work requires clearer and more robust justification. It appears that the approach is closely related to multi-agent reinforcement learning, where, in many cases, approaching a multi-agent system often involves adopting a central agent baseline. This central agent would process all incoming information and operate within an action space equivalent to the joint action space of the individual agents.

Explicit natural language communication could be helpful in human-vehicle settings (we cannot centralize humans in the multiagent system), but given the problem formulation in this work, as well as its potential applications in smart cities, one might question whether this specialized framework is necessary. It would be valuable for the authors to elaborate on why a decentralized approach is essential in this context.

### Weakness 2: Unclear method and training.
The so-called "training" in this work does not align with traditional machine learning training paradigms.
> For each LLM-based learning method, we train the models for up to 30 episodes per scenario, with early stopping if the scenario is solved, indicated by 10 consecutive successful episodes. After training, we evaluate each method over 30 episodes and report the average performance across these evaluations

However, on closer examination (as detailed in the appendix), this approach appears more akin to an agentic framework where the LLM is prompted to reason, propose actions, and reflect within episodic contexts, rather than engaging in conventional gradient-based learning. The overall experiment is also not systematic as "30 episodes" provides limited information about what has been used in testing and learning. The qualitative examples only show snippets of interactions.

This approach also contrasts with more complex agent-based vision-language models [5-7], which often account for a broader range of visual context and alignment issues. The simplicity of this framework raises concerns, particularly in overlooking these ambiguities.

### Weakness 3: Related work.
The authors could consider improving the related work.

> However, training such models requires extensive data. At the time of writing this paper, only a limited number of datasets exist that provide language commentary data for single-agent driving scenarios (Kim et al., 2018; 2019; Qian et al., 2023; Sima et al., 2023). To the best of our knowledge, datasets featuring natural language data for inter-vehicle communication are not yet available.

To the best of my knowledge, language datasets featuring human-vehicle communication [1-2] and vehicle-vehicle (multi-agent) collaboration [3-4] already exist. Although these resources do not diminish the novelty of this work, the authors should provide a detailed discussion on how this paper differs from previous efforts, rather than simply stating that similar datasets “are not yet available.”

[1] Deruyttere T, Vandenhende S, Grujicic D, Van Gool L, Moens MF. Talk2Car: Taking Control of Your Self-Driving Car. EMNLP 2019.

[2] Ma, Z., VanDerPloeg, B., Bara, C.P., Huang, Y., Kim, E.I., Gervits, F., Marge, M. and Chai, J., 2022, December. DOROTHIE: Spoken Dialogue for Handling Unexpected Situations in Interactive Autonomous Driving Agents. EMNLP Findings 2022.

[3] Suo S, Regalado S, Casas S, Urtasun R. Trafficsim: Learning to simulate realistic multi-agent behaviors. CVPR 2021.

[4] Wu W, Feng X, Gao Z, Kan Y. SMART: Scalable Multi-agent Real-time Simulation via Next-token Prediction. arXiv 2024.

[5] Mao J, Ye J, Qian Y, Pavone M, Wang Y. A language agent for autonomous driving. COLM 2024.

[6] Tian X, Gu J, Li B, Liu Y, Hu C, Wang Y, Zhan K, Jia P, Lang X, Zhao H. Drivevlm: The convergence of autonomous driving and large vision-language models. CoRL 2024.

[7] You J, Shi H, Jiang Z, Huang Z, Gan R, Wu K, Cheng X, Li X, Ran B. V2X-VLM: End-to-End V2X Cooperative Autonomous Driving Through Large Vision-Language Models. ArXiv 2024.

**Questions:**

Question 1:  Given the problem formulation in this work, as well as its potential applications in smart cities, why do we even need a decentralized approach like the one proposed? Would authors compare to the centralized agent as a baseline?

Question 2: How would visual information be handled in this work since CARLA is used for physical simulation?

Question 3: The action space seems to be high-level and discrete. Did the author rely on CARLA's built-in local motion planner, which has access to ground truth information about the environment? Is the focus of this work on the decision-making part rather than the actual control? (If so, I recommend changing the wording of "control policy")

---

> ### Author Response · Authors · 2024-11-22
> **Official Comments by Authors [1/2]**
>
> Dear Reviewer 9yf4,
>
> Thank you for your detailed review and constructive feedback. We sincerely appreciate the time and effort you have dedicated to evaluating our work and providing us with insightful comments. Below, we address your points and suggestions:
>
> ---
>
> **1. Decentralized vs. Centralized Control Setting**
> We acknowledge that the centralized control baseline could be included as an ablation study. For example, one research question could explore whether a single LLM agent with access to all information and control over all agents resolves the conflict among agents without communication. We will try to add an extra set of experiments to evaluate an LLM’s capability to solve the scenarios in negotiation scenarios, and if we get time during the rebuttal phase, we will include the report on whether reflection will improve LLM’s capability in resolving the scenarios. We would appreciate feedback from the reviewer on the ablation study setting.
> However, a centralized setting could nullify scenarios like Perception-Red-Light, where no vehicle would violate the red light under centralized control. We provide detailed reasoning in **Q1** and **Q2** of the **General Response** for why our research does not adopt a centralized control setting. Since the core research question in our paper is how to generate/understand natural language messages and incorporate the message into driving, this extra ablation study will be included in the appendix for reference.
>
> **2. Agent-based Framework and Explanation of “Training”**
>
> *The reason why we did not use a **gradient-based** method lies in*
> - [Line 92-95] No well-established dataset for V2V language communication, and the diverse conventions of cooperation cannot be easily modeled through behavior cloning [1];
> - [Line 248-251] we constrain the message space to be a natural language, but the gradient-based method can make LLM develop artificial language that is only comprehensible by agents that participate in the training [2];
> - has low sample efficiency, as observed in our preliminary experiments with RL;
> - are prone to overfitting specific scenarios and forgetting general knowledge.
> On the contrary, the agentic method serves as an effective initial step toward generating natural language messages as a baseline from which to improve. One can easily store the knowledge learned from debriefing and retrieve relevant knowledge when a similar scenario comes up.
>
> *The reason why we did not use a **VLM** lies in*
>
> The papers [5-7] are closely related works, but our argument is that we strip out the complexity of using a (cooperative vision language [7]) model to understand visual scenes, which is an independent research question, and highlight language message generation and understanding (detailed in **Q1** in General Response). On the other hand, our framework does not stop users from creating sensors for agents in the future, and users can easily replace LLM with VLM when the VLM gets stronger in spatial and temporal reasoning. According to our preliminary experimentation with VLMs (GPT-4o, LlaVA), they are pretty bad at even understanding the environment and traffic situations, and those foundation models are not yet able to incorporate more sensor modalities like multi-camera, LiDAR, IMU, etc to perceive the environment.
>
> *Explanation of **Training Method***
>
> Regarding the “training”, we acknowledge that our proposed method can be seen as an agentic method, and the “training” is realized by adapting the context/knowledge prompted to language models. We will modify the description of the framework/method in the next revision. We demonstrate the learning/adapting process in Figure 2 and Algorithm 1 in Appendix. The following are explanations of learning with code: In a training [[train_centralized_debrif.py]](https://anonymous.4open.science/r/talking-vehicles/code/train/train_centralized_llm.py), we first rollout an episode [[rollout.py]](https://anonymous.4open.science/r/talking-vehicles/code/utils/rollout.py) so that we get feedback on the episode from the scenario evaluator [[evaluator.py]](https://anonymous.4open.science/r/talking-vehicles/code/envs/utils/evaluator.py), and we retrospectively re-label all the (state, action, next state, other agent’s reaction, feedbacks) transitions in the replay buffer of each policy [[process_learning_data()]](https://anonymous.4open.science/r/talking-vehicles/code/utils/rollout.py). The debrief process is handled by a debrief manager (or a host for discussion) [[debrief_manager.py]](https://anonymous.4open.science/r/talking-vehicles/code/policies/debrief_manager.py), and the discussion context is sampled from the replay buffer of each agent to join the discussion, and dialog is then summarized and stored by the policies after reflection [[reflection.py]](https://anonymous.4open.science/r/talking-vehicles/code/policies/utils/reflection.py).

---

> ### Author Response · Authors · 2024-11-22
> **Official Comments by Authors [2/2]**
>
> The policies used are all LLM_policies [[llm_policy.py]](https://anonymous.4open.science/r/talking-vehicles/code/policies/llm_policy.py) with Llama or OpenAI interface.
>
> **3. CARLA Visual Information and Command/Control/Planning**
>
> **3.1 Visual Information**
> In our simulation framework, we dedicated significant effort to developing a partial observation captioner [[partially_observable_captioner.py]](https://anonymous.4open.science/r/talking-vehicles/code/utils/partial_observable_captioner.py), which translates simulator information into text descriptions as accurately as possible. This modularized component provides a detailed representation of the environment, including vehicle location, speed, lane markings, and descriptions of **visible** [[shapley_geometry.py]](https://anonymous.4open.science/r/talking-vehicles/code/utils/shapely_geometry.py) surrounding vehicles with their states by directly accessing simulator data. By using the captioner, users can bypass the need to process raw sensor data, allowing them to focus on the core research question (as discussed in **General Response Q1**).
>
> **3.2 Command/Control/Planning**
> The main interface between the policies and the environment is through the agent [[comm_agent.py]](https://anonymous.4open.science/r/talking-vehicles/code/agent/comm_agent.py). At the beginning of an episode, a global planner plans a sequence of reference waypoints for an agent to follow. To enable the execution of agent commands, we implemented an atomic command follower [[atomic_command_follower.py]](https://anonymous.4open.science/r/talking-vehicles/code/agent/atomic_command_follower.py), which translates high-level commands into actionable (1) short-term path-planning tasks and connects back to the global reference route, such as "change to the left lane." and (2) changes in throttle, brake for speed control. A low-level local planner [[local_planner.py]](https://anonymous.4open.science/r/talking-vehicles/code/agent/local_planner.py) will plan vehicle controls through the waypoints and PID controllers.
>
> **4. Related work**
> Thank you for sharing all the related work! We will incorporate this related work in our next revision. Nonetheless, the new related work does not undermine our originality and novelty.
> Although [1-2] features human-vehicle communication and instruction, and [3-4,7] discusses V2V collaboration, we differ and relate to the two streams of the research in that we explore the possibility of V2V/V2X/Vehicle-Human multi-agent collaboration (not only perception but also negotiation) in the natural language message space. Our work also differs from the LLM-based driving study [5-6] by emphasizing multi-vehicle interactions and communications.
>
> ---
>
> We hope that our responses have addressed your concerns and provided additional clarity on the key contributions, methodologies, and motivations of our work. If there are any remaining questions or areas where further explanation is needed, we would be more than happy to provide additional details or conduct further experiments as necessary.
>
> ### References
> [1] Cui, Brandon, et al. "Adversarial diversity in hanabi." The Eleventh International Conference on Learning Representations. 2023.
>
> [2] Guo, S., Ren, Y., Mathewson, K., Kirby, S., Albrecht, S. V., & Smith, K. (2021).
> Expressivity of Emergent Languages is a Trade-off between Contextual Complexity and Unpredictability, International Conference on Learning Representations, 2022

---

> > ### Comment · Reviewer_9yf4 · 2024-11-26
> >
> > Thank you for your response.
> >
> > ---
> >
> > >  A low-level local planner [local_planner.py] will plan vehicle controls through the waypoints and PID controllers.
> >
> > I read your code. If the `class LocalPlanner(object)` is employed in low-level planning, it should be acknowledged that this inherently provides lane-following capabilities by utilizing ground truth information from the simulation instead of relying on sensor data. Consequently, all contributions in this context are limited to high-level decision-making. This is fine but it needs to be clear.
> >
> > ---
> >
> > The authors suggest that a centralized control setting could effectively handle scenarios like Perception-Red-Light, where no vehicle would violate the red light. If a centralized system is capable of resolving this issue naturally, it would be helpful to clarify why the proposed decentralized approach, which introduces the additional complexity of natural language communication, is necessary for addressing such a scenario. Without a clear justification, **this may create an artificial problem that may not align with practical, real-world needs.**
> >
> > ---
> >
> > The authors seem to conflate interpretability with the use of natural language messages, suggesting that V2V or multi-agent collaboration must rely on natural language for interpretability. However, interpretability does not necessarily require natural language. Many systems achieve interpretability effectively through structured formats or symbolic representations, which may be more appropriate and efficient for V2V communication.
> >
> > ---
> >
> > > Since the core research question in our paper is how to generate/understand natural language messages and incorporate the message into driving ...
> >
> > If this is the core research question, then I don't understand why not directly study human-vehicle communication, which has established datasets and tasks. This may provide more immediate insights into the application of natural language in driving scenarios, rather than this long and confusing logic chain.
> >
> > ---
> >
> > > Although [1-2] features human-vehicle communication and instruction, and [3-4,7] discusses V2V collaboration, we differ and relate to the two streams of the research in that we explore the possibility of V2V/V2X/Vehicle-Human multi-agent collaboration (not only perception but also negotiation) in the natural language message space. Our work also differs from the LLM-based driving study [5-6] by emphasizing multi-vehicle interactions and communications.
> >
> > Simply emphasizing the difference from existing work does not necessarily validate the premise that the problem itself is essential. It would be beneficial to further justify why this particular approach (multi-agent collaboration through natural language, done by prompting-based LLM agents) addresses a gap or need that is not already covered by existing research.
> >
> > ---
> >
> > I am not planning to make changes to the ratings at this point, but I encourage the authors to further justify the motivations of this work.

---

> ### Author Response · Authors · 2024-12-02
>
> We sincerely thank you for your valuable feedback and questions. We have incorporated your suggestions regarding writing clarity and scoping the contribution in the high-level commands in the revised paper. We provide further motivation for the research and support for the problem here (in addition to the introduction and related work)
>
> - The deployment of autonomous vehicles is a gradual process, and privacy/security issues posed by centralization control make it infeasible to reach full centralization in autonomous vehicle control immediately. The alternative setting we consider where our method/task could possibly be useful is a mixed-autonomy setting, where autonomous vehicles and human vehicles coexist. There is no assumption about the control over human-driven vehicles or their rationality. Human drivers may generally behave rationally but can exhibit aggressive behaviors, such as running a red light. This line of research is well documented by a large body of studies, representative works include [1-2]. Our simulation framework supports human interaction with autonomous vehicles, which we aim to explore as a future research direction.
> - Given the first argument above, natural language is a promising interface to bridge human and vehicle communication, including inter-vehicle and in-vehicle communication. The reviewer mentioned in-vehicle (human-vehicle) communication as a solid research topic. However, we argue that inter-vehicle communication is also a solid research topic and is seriously considered by the real industry [3-4]. We believe it is a natural extension to study the possibility of inter-vehicle communication in natural language.
> - Compared to many prior works in V2V or V2X that promote early fusion, representation fusion, or late fusion in the perception module[5], natural language enjoys better interpretability by layman humans in terms of the generated message and enables the ad hoc coordination among vehicles through language instructions. For example, there is no clear signal for a private sedan to inform a trailing vehicle about its backup plan to enter a parking spot behind it; or even if the emergency vehicle alarms loudly, many human drivers will not realize it is the emergency vehicle and give way to it accordingly, which could be improved by information sharing on the emergency vehicles. The reviewer's concern might also be about explainability regarding decision-making or automated reasoning processes, but the problem that we explore does not prevent the use of a symbolic method for decision-making and its combination with LLMs to generate natural language messages.
> - Our methodology builds on related work in prompt-based agentic methods for driving, as detailed in the related work section. And the research gap is that most of them do not consider multi-agent interactions and the closest work to our work (we compared with it with adaptation to our task and environment) does not optimize communication messages [6]. Our research explores the potential for LLM agents to acquire collaborative knowledge through multi-agent interactions without relying on human-labeled data.
> Our work aligns with the recent interest in exploring language interaction games and self-improvement [7], but we ground the exploration in the autonomous driving context.
>
> We appreciate your time discussing this work with us, hope our explanation provide a better explanation on the motivation.
>
> ----
>
> [1] Wu, C., Kreidieh, A. R., Parvate, K., Vinitsky, E., & Bayen, A. M. (2021). Flow: A modular learning framework for mixed autonomy traffic. IEEE Transactions on Robotics, 38(2), 1270-1286.
>
> [2] Kazemkhani, S., Pandya, A., Cornelisse, D., Shacklett, B., & Vinitsky, E. (2024). GPUDrive: Data-driven, multi-agent driving simulation at 1 million FPS. arXiv preprint arXiv:2408.01584.
>
> [3] L. Gallo and J. Harri. Short paper: A lte-direct broadcast mechanism for periodic vehicular safety communications. In IEEE Vehicular Networking Conference 2013, pages 166–169. IEEE, Dec 2013. doi: 10.1109/VNC.2013.6737604.
>
> [4] Qualcomm. Lte direct proximity services, 2019. URL https://www.qualcomm.com/invention/ technologies/lte/direct
>
> [5] Wang, T. H., Manivasagam, S., Liang, M., Yang, B., Zeng, W., & Urtasun, R. (2020). V2vnet: Vehicle-to-vehicle communication for joint perception and prediction. In Computer Vision–ECCV 2020: 16th European Conference, Glasgow, UK, August 23–28, 2020, Proceedings, Part II 16 (pp. 605-621). Springer International Publishing.
>
> [6] Hu, S., Fang, Z., Fang, Z., Deng, Y., Chen, X., & Fang, Y. (2024). Agentscodriver: Large language model empowered collaborative driving with lifelong learning. arXiv preprint arXiv:2404.06345.
>
> [7] Schaul, T. (2024). Boundless Socratic Learning with Language Games. arXiv preprint arXiv:2411.16905.

---

> > ### Comment · Reviewer_9yf4 · 2024-12-02
> >
> > It seems there Is some concept change here. LTE (Long Term Evolution) typically refers to wireless data transmission [3-4], so it is not immediately clear why inter-vehicle communication would involve natural language, emergent latent communication is more than enough. If the motivation is to improve interpretability, it would be helpful to discuss how reliable language-based explanations are and whether they faithfully reflect the internal representations of LLMs. Additionally, the paper does not appear to address or justify this specific choice of setting, which would benefit from further elaboration.

---

> ### Author Response · Authors · 2024-12-02
>
> Thank you for your prompt response. Below we provide clarifications and detailed responses to your questions.
> 1. **Clarification on [3-4]**
> >It seems there Is some concept change here. LTE (Long Term Evolution) typically refers to wireless data transmission [3-4], so it is not immediately clear why inter-vehicle communication would involve natural language
>
> References [3-4] are supporting evidence for the argument made about *inter-vehicle communication is considered by the industry and academia.* LTE is the basis of C-V2X(Cellular Vehicle-to-Everything)[4, https://en.wikipedia.org/wiki/Cellular_V2X ] and the message in [3] is the using LTE for vehicular communication.
> Leading telecom providers, including but not limited to Qualcomm, promote on-board vehicular communication support.
> - https://academy.qualcomm.com/course-catalog/Introduction-to-C-V2X
> - https://www.qualcomm.com/products/automotive/connectivity-positioning
> - https://www.rcrwireless.com/20240819/policy/dot-releases-national-v2x-plan
>
> 2. **Latent Communication and Its Limitations**:
> >...why natural language, emergent latent communication is more than enough.
>
> Latent communication, while valuable, is insufficient for mixed-autonomy scenarios involving human drivers. Human interpretable communication remains crucial in such contexts. Future work could explore the safe usage and translation of V2X communication standards (exemplified in https://www.sae.org/standards/content/j2735_202409/ ) into human-interpretable formats.
> Although latent representations can infer agents' intentions for coordination and negotiation (e.g., as demonstrated by LILI [8]), they present challenges in generalization and interpretability. Questions arise, such as:
> -  What happens when other vehicles utilize different encoders from training for latent representations? (aka, speak different languages, compared to estabalished translation among natural languages)
> -  How can these representations be explicitly and robustly explained?
> -  How effectively do learned latent representations generalize across diverse scenarios?
>
> Natural languages, though sometimes vague, can convey clear intentions through multiple message exchanges to avoid vague expressions. Future research could learn how to avoid ambiguity while keeping the message concise in natural language. We envision this work, which allows agents to interact with each other in the simulator, as a potential place to discover and improve efficient natural language protocols.
>
> 3. **The faithfulness of Generated Messages**:
> > If the motivation is to improve interpretability, it would be helpful to discuss how reliable language-based explanations are and whether they faithfully reflect the internal representations of LLMs.
>
> Ensuring that generated messages faithfully reflect the internal representations of LLMs is an open research question. AI alignment, validation tools such as **probing** techniques, or direct observation of latent states are needed to evaluate this fidelity effectively.
>
> 4. **Choice of Scenarios**:
> >the paper does not appear to address or justify this specific choice of setting, which would benefit from further elaboration.
>
> We assume that the reviewers are referring to the scenario setup in our paper rather than the mixed-autonomy context (please see Argument 1 in our previous response). Our scenarios were inspired by accident-prone situations in CARLA's leaderboard https://leaderboard.carla.org/scenarios/. We extended these scenarios to accommodate **multiple learning agents** and designed additional settings that may benefit from communications, including Negotiation-Overtake, Negotiation-Highway-Exit, and Negotiation-Highway-Merge (Explanation of scenarios in Table 4 in Appendix B).
>
> 5. Finally, we emphasize that our research is a study on multi-agent interaction and self-improvement of LLM agents with natural language communication. This work is contextualized within autonomous driving but has broader implications for improving natural language protocols and multi-agent coordination. Related scenarios include but are not limited to human-robot soccer, AI accompaniment in language board games, and ad hoc human-AI coordination in emergency rescue and etc.
>
> We hope this explanation is to the point. Please let us know if further elaboration is required.
>
> ----
>
> [8] Xie, A., Losey, D., Tolsma, R., Finn, C., & Sadigh, D. (2021, October). Learning latent representations to influence multi-agent interaction. In Conference on robot learning (pp. 575-588). PMLR.

---

### Official Review · Reviewer_7Vjf · 2024-11-03

**Soundness:** 3
**Presentation:** 3
**Contribution:** 2
**Rating:** 3
**Confidence:** 4

**Summary:**

The paper proposes the use of natural language for communication between vehicles as a way to improve autonomous driving. The paper focuses on specific driving tasks (overtake, red light, left turn, highway exit, highway merge) and presents a new method, called LLM+Debrief, to learn how to generate messages to other vehicles and generate related control policies. The experimental results included have been obtained in a gym-like simulation environment built on the CARLA urban driving simulator. The experiments address in-episode communication, chain-of-thought reasoning, and post-episode debriefing. The results show that communication facilitate cooperation and that reflection and debriefing improve the performance when the vehicles negotiate with each other. The problems are modeled as "a multi-agent partially observable and general-sum game".  Each vehicle is assumed to be cooperative. The objective of each vehicle is to optimize the time to reach its destination. The main novelty is to provide autonomous vehicles with the ability to interact with each other in natural language, which facilitates the interactions with other vehicles.

**Strengths:**

The idea proposed is interesting and innovative. It seems premature, but it opens a new direction of work. The paper has value as a basic feasibility study.

The paper provides positive experimental results, which have been obtained in simulation, on the value of having vehicles communicate and negotiate with each other.

The writing is clear and the structure of the paper is well organized.

**Weaknesses:**

A question that comes to mind is why using natural language for vehicles to communicate is better that using an artificial communication language. The paper says that natural language could also allow human drivers to participate in the conversation, but it might also distract the drivers.

Each vehicle generates observation messages and decides driving plans in collaboration with the other vehicles. From the examples shown in the figures, there could be a lot of messages, sometimes too many.  The figures shown do not have many vehicles, at most 3 or 4, but there could be many more, for example in city traffic on streets with multiple lanes.

In addition to deciding what to do, each vehicle has to generate its control policy.  An issue is the time needed to share messages and what happens if there is message congestion. In the simulation experiences are collected every 0.5 seconds and kept for 2 seconds.  There is no information on what would the timing should be in real vehicles. Also no information is given about the time needed to generate the control policy.  In the simulation, are the steps for all the vehicles synchronized with the same discrete time steps?

The information sensed by the vehicle is translated into text, since it has to be used in the conversation. No indication is given about the time this will take.

The assumption of truthful information and collaborative attitude of all the vehicles is very strong and might not correspond to reality.

Minor issue: the paper says that each vehicle can express its individual preference for its objective, but does not show any examples.  All preferences are assumed to be the same. i.e., optimize the time to reach its destination.

**Questions:**

Please clarify some of the points listed as weaknesses, for example by providing some information on the temporal issues.  The system will have to work in real-time.

Please also address the potential for message congestion when there are many vehicles in close proximity.

Finally, please address the issue of how to recognize untruthful information or address aggressive driving.

---

> ### Author Response · Authors · 2024-11-22
> **Official Comments by Authors [1/2]**
>
> Dear Reviewer 7Vjf,
>
> We greatly appreciate your valuable feedback and the opportunity to address your concerns and clarify our work. Below, we provide detailed responses to each of your concerns.
>
> ---
>
> **1. Natural Language Communication vs. Artificial Language and Individual Preferences**
> We investigate natural language as a communication medium for the following reasons:
> 1. It opens opportunities for vehicles to communicate with humans with understandable language without specific training on humans (truck drivers and pilots have developed an abbreviated language system but are usually non-interpretable by everyday urban drivers).  Regarding the concern of distracting human drivers, we envision that communicating natural language messages through a vocal interface will not distract human drivers too much (e.g., we have gotten used to listening to the radio and talking while driving), especially when the human drivers are merely serving as co-pilots of autonomous driving systems. It would be an interesting future study to investigate the ideal communication interface through human-in-the-loop experiments.
> 2. It extends the usage of V2V communication from cooperative perception to negotiation. Direct driving intentions or negotiation helps handle corner cases for autonomous driving; for example, two cars could have negotiated a plan in a narrow hallway cases like this discussion on X: https://x.com/j_foerst/status/1811676036418937208
> 3. It makes the communication explicit and increases the interpretability for monitoring and intervention.
> 4. The proposed method with an LLM can execute **language-specified goals and preferences** (such as “Exit highway via the leftmost lane and follow the exit ramp, prioritizing safe speed while in a hurry.” [[code](https://anonymous.4open.science/r/talking-vehicles/code/envs/scenarios/multiagent_scenarios/highway_exit/highway_exit__negotiation_comm_risky.yaml)]) and provide reasoning for decision-making, enhancing the explainability of autonomous vehicles' driving decisions and controllability to avoid congestion created by autonomous vehicles getting confused by the situation.
>
> **2. Managing Message Volume and Congestion**
> We agree that an increased number of vehicles could lead to message congestion, especially in urban scenarios. In our current setup, we included a few vehicles to assess the feasibility of natural language-based communication, and the message dialog could be kept longer as the context window of LLM becomes larger (GPT-4o has a 128K context window, while Llama3 has an 8k context window). For future studies, we plan to explore message prioritization strategies and filtering methods that allow only critical messages to be sent in high-traffic scenarios. Furthermore, our simulation synchronizes agent decision steps (to save some language model inference time [[code]](https://anonymous.4open.science/r/talking-vehicles/code/utils/rollout.py)), but this constraint can be easily relaxed in the future with lighter models. Still, this synchronization may need refinement in a real-world setting, where asynchronous messaging protocols or decentralized architectures might better address message congestion and scalability challenges. Another mark about the message size: although we did not strictly constrain the message size, the language model messages were all very concise, as demonstrated in **Table 3** in **General Response Q3** and qualitatively by [[videos]](https://anonymous.4open.science/r/talking-vehicles/video); the messages are on the top left of the videos.

---

> ### Author Response · Authors · 2024-11-22
> **Official Comments by Authors [2/2]**
>
> **3. Temporal Issues and Captioner/Control/Message Generation Latency**
> In our simulation, agents make decisions at 0.5-second intervals (every 10 frames), which is similar to human driving response times, and the simulation is synchronized with a frame rate of 20 fps. Control commands and message generation are generated at the same time through large language models, which take 15-20 seconds (**Table 3 in General Response**) in the real world to create both reasoning and decisions (command, decisions). For practical consideration, we will explore internalizing the experience from debriefing into smaller language models (please check the **Q3** in the General Response) to meet the real-time standard, but this is left to separate future work. Atomic planners that execute the control commands to plan trajectory and speed [[atomic_command_follower.py]](https://anonymous.4open.science/r/talking-vehicles/code/agent/atomic_command_follower.py) and captioners [[partially_observable_captioner.py]](https://anonymous.4open.science/r/talking-vehicles/code/utils/partial_observable_captioner.py) that translate simulator information to texts consume very little time (Table 3).
>
> **4. Assumption of Truthful Information and Collaborative Attitudes**
> We acknowledge a truthful and cooperative attitude is a strong assumption in real-world driving settings, and we only uphold this assumption in this work [stated in Line 239], following the training assumption in [1]. However, we recognize the importance of exploring diverse behaviors demonstrated by humans or agents where their messages could be deceptive or adversarial, and we plan to address this in future work. Adversarial attacks in the vehicle-to-vehicle (V2V) communication system [2] and attacks on language models are active research areas as well. While beyond the scope of this research, addressing challenges such as language model hallucination and the generation of deceptive messages will be crucial for advancing human-AI collaboration.
>
> ---
>
> If all concerns have been adequately addressed in our responses and the revised manuscript, we kindly request that you consider revising your evaluation of the paper. Please let us know if there are any remaining questions or additional concerns we can further address.
>
> Thank you once again for your thoughtful review and consideration.
> ### References
> [1] Meta Fundamental AI Research Diplomacy Team (FAIR)†, Bakhtin A, Brown N, et al. Human-level play in the game of Diplomacy by combining language models with strategic reasoning[J]. Science, 2022, 378(6624): 1067-1074.
>
> [2] Zhang, Q., Jin, S., Zhu, R., Sun, J., Zhang, X., Chen, Q. A., & Mao, Z. M. (2024). On data fabrication in collaborative vehicular perception: Attacks and countermeasures. In 33rd USENIX Security Symposium (USENIX Security 24) (pp. 6309-6326).
>
> [3] Wen, L., Fu, D., Li, X., Cai, X., Ma, T., Cai, P., ... & Qiao, Y. Dilu: A knowledge-driven approach to autonomous driving with large language models. The Twelfth International Conference on Learning Representations. 2024.

---

> ### Comment · Reviewer_7Vjf · 2024-11-27
>
> Thanks for the explanation. While I understand your interest in using natural language to communicate, I am concerned about the congestion of messages, which should have been addressed. Thanks also for recognizing the importance of addressing false or adversarial information to create a viable system.

---

> > ### Author Response · Authors · 2024-12-02
> >
> > We agree that message congestion is a concern when deploying the method in the real world. Below, we address the concern by first demonstrating that message congestion is not a concern from the communication system perspective while highlighting that the primary challenge lies in the agents’ self-identification with an increasing number of message participants.
> >
> > First, we believe modern onboard communication systems are well-equipped to filter messages and allocate communication slots to avoid congestion and collisions (e.g., [1]). The size of the language messages is not a major concern, as the generated messages (<0.1 Mbps, as reported in Table 3 of the rebuttal) are well below the throughput capacities of DSRC (2 Mbps) [2] or C-V2X (7.2 Mbps) [3-4].
> >
> > However, the main limitation we encountered in scaling experiments with more communicative agents stems from the LLM agents' ability to identify themselves within lengthy message dialogs. While stronger LLMs can partially mitigate this issue, it remains an area for improvement. We believe that agents' capacity to process more information scales with advancements in LLM capabilities. Exploring the inclusion of additional agents in coordination tasks and further scaling the methodology presents an exciting direction for future research.
> >
> > Thank you again for your valuable feedback and raise interesting discussion questions.
> >
> > ----
> > [1] Ashraf, M. I., Bennis, M., Perfecto, C., & Saad, W. (2016, December). Dynamic proximity-aware resource allocation in vehicle-to-vehicle (V2V) communications. In 2016 IEEE Globecom Workshops (GC Wkshps) (pp. 1-6). IEEE.
> >
> > [2] J. B. Kenney. Dedicated short-range communications (dsrc) standards in the united states. Proceedings of the IEEE, 99(7):1162–1182, July 2011. ISSN 0018- 9219. doi: 10.1109/JPROC.2011.2132790.
> >
> > [3] L. Gallo and J. Harri. Short paper: A lte-direct broadcast mechanism for periodic vehicular safety communications. In IEEE Vehicular Networking Conference 2013, pages 166–169. IEEE, Dec 2013. doi: 10.1109/VNC.2013.6737604.
> >
> > [4] Qualcomm. Lte direct proximity services, 2019. URL https://www.qualcomm.com/invention/ technologies/lte/direct

---

### Official Review · Reviewer_EnFV · 2024-11-04

**Soundness:** 3
**Presentation:** 3
**Contribution:** 2
**Rating:** 6
**Confidence:** 4

**Summary:**

The authors propose a method LLM+Debrief policy that deals with multi-agent interactions tailored to traffic environments. Each agent is a vehicle with a goal and communicates in natural language. This goal, along with environmental observations and previous knowledge, outputs a message to neighbors and an action. The episodes progress as each agent outputs messages and actions at every step. At the end of the episode, agents receive a summary of their performance from the environment. The agents are then involved in a debriefing session where they share reasoning and discuss strategies that provide feedback on the LLM policy.

The method is evaluated in a CARLA simulator with multiple baselines, including a non-language baseline with and without communication. Results are presented for various levels of communication via reflection and debriefing.

**Strengths:**

The study provides an interesting approach to enable autonomous vehicles to interact with each other in various driving scenarios. The authors have developed a versatile TalkingVehiclesGym that uses natural language and incorporates partial observability. The framework can serve as a tool for natural language-based cooperative driving scenarios.

The work demonstrates the benefits of leveraging knowledge gained so far along with communication to improve cooperation over silent methods.

The introduction of post-episode debriefing sessions is a significant addition.

The method is tested in the CARLA simulator with a comprehensive set of baselines, including scenarios both with and without communication. The methodology is explained well. The figures aid in the understanding of the paper.

**Weaknesses:**

The paper is generally well-written and organized; however, there are areas where clarity could be improved. More explanation is needed on how the reward structure is utilized.

The results could also include more evaluation details in a tabular form. For example, these could include the time to evaluate and the real-time equivalent processing time to show how this method could work in a real-world setting.

Providing more implementation details could also explain the choice of the number of episodes per scenario, which seems to be low. Could the authors justify their choice of episode number or explain how they determined this was sufficient for their experiments?

**Questions:**

A) Some of the supplementary materials have detailed the simulation times/elapsed times. Are these representative of a similar real-world implementation? Could the authors provide a table with columns for each method, showing evaluation time and real-time equivalent processing time for each scenario tested? Additionally, metrics could include message generation time and decision-making latency.

B) Can this LLM+Debrief include the Coopernaut method? Since this provides a relatively better performance, it would be interesting to see if combining the two would improve LLM+Debrief.

Minor comments:
1. Page 6: In line 305, “pushlish” → publish?
2. Page 6, lines 315-318 have a repeated sentence.

---

> ### Author Response · Authors · 2024-11-22
>
> Dear Reviewer EnFV,
>
> Thank you very much for your thorough review and thoughtful feedback on our work! We appreciate your insights and address each of your questions below.
>
> ---
> **1. Real-world Evaluation and Processing Time**
> We understand the importance of real-world applicability. The elapsed times shown in our videos reflect game-world time. For real-world implementation, processing time would vary based on factors like agent count, model selection (GPT is generally faster than Llama3-8B on our A100 machines), and whether the “episode” is “terminated” early. We parallelize agent inference across multiple GPUs, which reduces latency significantly. The chain-of-thought prompting inference speed ranges between 10-20 seconds per decision. To provide clear insights into the overall latency and real-world feasibility, we’ll measure the latency values more precisely and add a table in the appendix with the average model reasoning latency and decision/message generation latency.
>
> We acknowledge this work is a proof-of-concept exploratory study of natural language vehicle-to-vehicle communication in autonomous driving. As with many prior works exploring LLMs for autonomous driving, the significant computational demands of foundation models pose challenges and hinder scalable real-time applications. Nevertheless, fitting small language models on resource-limited edge devices is an active research field. In future work, we look forward to taking advantage of the latest progress in this research direction to internalize the experience from debriefing into smaller language models to meet the real-time standard. Please check our response to **Q3** in the **General Response** for more details.
>
> **Table 3** in the **General Response** summarizes the average latencies and message sizes for each scenario under the communication setting, evaluated using Llama3-8B-Instruct on Nvidia A100 GPUs and Intel Gen 10 CPUs. The metrics include partial observable captioner latency (in seconds), reasoning latency (in seconds), decision latency (in seconds, excluding reasoning latency), and message size (in Mb). Data is aggregated over 10 episodes at each LLM decision step. Notably, GPT-4o online APIs demonstrate 2x faster generation speeds (\~16 seconds vs \~8 seconds). Scenarios without communication exhibit slightly lower reasoning and decision latencies compared to those with communication (\~16 seconds vs \~13 seconds), though the differences are within the same order of magnitude.
>
> **2. Integration with Coopernaut**
> Integrating Coopernaut with LLM+Debrief presents an interesting possibility. Coopernaut’s strength lies in learning which LiDAR-based representations to share. However, the senders and receivers must be trained on the same representation a priori. Also, these LiDAR-based representations are not directly interpretable to humans, preventing their applications in mixed autonomy with human drivers involved. As the LLM approach matures, Coopernaut could potentially be adapted to produce natural language messages, directly imitating LLM policies—a development we find quite exciting! Until then, Coopernaut could be reserved for interactions among all autonomous vehicles, while the approach introduced in this paper moves towards enabling human drivers to participate.  Exploring this integration in the future could enhance interpretability and cooperation, bridging perceptual data and language-driven interactions.
>
> **3. Reward Structures and Implementation Details**
> We refer the reviewer to Q2 in General Response for the reward structure and how the task and preferences are specified for each agent. In short, each agent has a language-specified task and preference, such as “Exit highway via the leftmost lane and follow the exit ramp, prioritizing safe speed while in a hurry.” [[code]](https://anonymous.4open.science/r/talking-vehicles/code/envs/scenarios/multiagent_scenarios/highway_exit/highway_exit__negotiation_comm_risky.yaml)
> The choice of training episodes is primarily influenced by the limitations and capabilities of the context adaptation methodology and the language model used. Our findings indicate that the language model captures most of the key takeaways within a few episodes of interaction. However, beyond these episodes, the knowledge and context tend to stabilize, showing minimal changes thereafter
>
> ---
> Please let us know if there are any additional areas where we can provide further clarification. Thank you once again for your valuable feedback and for acknowledging our work!

---

> ### Comment · Reviewer_EnFV · 2024-11-26
>
> Thank you for your response. There were still some things missing in the author's comment, such as clarification of the choice of the number of episodes per scenario, which seems to be low. Could the authors justify their choice of episode number or explain how they determined this was sufficient for their experiments?
>
> The training implementation and details were only clear after reading the author's response #2 in [reviewer 7Vjf comment](https://openreview.net/forum?id=VYlfoA8I6A&noteId=qT9F00OM8c) and #2 in [reviewer 9yf4 comment](https://openreview.net/forum?id=VYlfoA8I6A&noteId=dD32dfF5ev).
>
> Furthermore, other reviewers also raised the practicality of natural language for V2V communication in real-life scenarios, which I had not fully considered earlier. Upon further reflection, I have decided to adjust my score from 8 to 6.

---

> > ### Author Response · Authors · 2024-12-02
> >
> > Thank you for your valuable feedback. We have incorporated your suggestions for clarifying the core concepts and clarification on learning methodology in the revised version of the paper.
> >
> > We are happy to discuss our method designs and are open to suggestions. Regarding the choice of the number of episodes per scenario, our primary consideration of the design lies in that
> > - The rich information provided by the environment feedback and reasoning in the learning phase. As humans, we can learn from a failure case pretty efficiently by reasoning, proposing new knowledge, and verifying if it works. Such a way of learning consumes significantly fewer examples than gradient-based reinforcement learning with function approximators. In autoregressive models, changing the knowledge in the form of context or prompt tokens can be seen as a rough way of changing parameters in the function approximators.
> > - Our early stopping mechanism (stop learning at 10 consecutive successful episodes) for learning indicates that if a learning process has been effective, it takes fewer than 30 episodes, often around 15 episodes.
> > - Conversely, if the learning fails, it is not easy to recover from a hallucinated knowledge even with more episodes. This issue arises due to hallucinated causal relations between policy (knowledge context) and consequences, which we observed with the large language model we used (Llama3-8B). However, this phenomenon improves with models like GPT-4o. We expanded the discussion on this limitation, including selecting what example to learn from and the difficulty in knowledge preservation over learning issues in Appendix E. We will explain this issue more in the main content of the paper in the next revision.
> >
> > Regarding the real-world application of natural language for V2V communication, we see strong motivations for adopting human-compatible language for the mix-autonomy (both human and autonomous vehicles exist) situations in the near future. LLM assistants are naturally able to parse received long natural language messages to concise phrases or instructions for autonomous vehicles or human driving while sitting inside.
> >
> > Our primary focus is testing and verifying the ability of LLMs to establish collaboration through communication and reason from others’ perspectives through interactions with themselves. For example, in the `Perception-Left-Turn` scenario, a human might easily say, “Hey, there’s a car going in the opposite direction while you’re turning left.” In contrast, LLM agents tend to observe details like, “There is a northbound vehicle turning left at the intersection; I am facing southbound; there is a vehicle in my right lane traveling southbound,” but may not recognize the necessity of sharing critical road information. Notably, the agent responsible for sharing road information (e.g., a truck) successfully identified the need to communicate information about southbound vehicles to the left-turning car, as detailed in Appendix F.2.
> >
> > Additionally, the latency of the framework using LLMs could be addressed by internalizing the thought process (or retrieving related reasoning) and training a small imitator to learn from large models.
> >
> > We understand your concern and respect your decisions. Thank you for your time and thoughtful consideration.

---

### Author Response · Authors · 2024-11-22
**General Response [1/3]**

We sincerely thank all reviewers for their valuable time and constructive feedback! We are encouraged that all reviewers found the paper interesting and contributive in terms of ideas, methodology, and/or simulation framework. We acknowledge that this is a proof-of-concept exploratory paper and agree that it could benefit from improved clarity and expanded discussions of its limitations. To address these points, we will submit a revised version by the end of the rebuttal phase, incorporating all discussion points raised. Additionally, we will update an anonymous GitHub repository with the sanitized code during this period.

In this general response, we provide justifications for this paper being a valuable contribution to ICLR and address key questions raised by the reviewers. Specific responses to each reviewer will be posted under their respective comments.

- A code folder and videos were submitted as supplementary materials along with the paper; references to them will follow the format `[code->module]` or `[video]`.
- Our anonymous repo is here https://anonymous.4open.science/r/talking-vehicles, which also contains `[code->module]` or `[video]` we will keep cleaning up the README and code.

**Q1: Why we believe this paper a valuable contribution to ICLR?**
- The problem we proposed is rooted in the intersection of multi-agent language games (e.g., Diplomacy, Werewolf) and autonomous driving, and the key aspect of the research problem is **“How can we enable autonomous agents to generate and understand natural language messages to facilitate cooperation and make movement decisions that incorporate these messages.”** The autonomous driving setting brings the multi-agent communication problem to a real-life setting and opens up challenging problems such as spatial and temporal understanding of the situation, as well as generating cooperative messages, which are beneficial for safe and effective driving. Furthermore, incorporating LLMs into driving is a promising direction for future research as it enhances the explainability by making the driving decisions more transparent, improves the interpretability of the communication between agents, and fosters greater interactivity with humans. This aligns with the trajectory of recent work in the field, as highlighted in our Related Work section.

- *Our contributed gymnasium-based simulation framework has significance and novelty for the community* as (1) the previous CARLA-based multi-agent gym [MACAD](https://github.com/praveen-palanisamy/macad-gym) does not fully support **real language/representation communication** and is less flexible for configuring different policies and agents. In contrast, ours has a carefully designed communication framework `code->comm/` and API structure and is easy to extend with more agents and more scenarios `code->envs->scenarios->multiagent_scenarios/` (2) Compared to [HighwayEnv](https://github.com/Farama-Foundation/HighwayEnv), our framework is designed for **intensive, partially observable** multi-agent interaction and communication with high-fidelity simulation and allows future usage of sensors and human interactions. (3) Our framework shares the fundamental multi-agent structure with [Concordia](https://deepmind.google/research/publications/64717/) and [MeltingPot](https://github.com/google-deepmind/meltingpot) to support diverse behaviors and general-sum games, but it introduces a completely different setting in autonomous driving.

- *Our proposed method and improvement in the methodology present the most direct and novel first attempt to approach the outlined problem*. Unlike Cicero [1], which benefits from **expert datasets** and behavior cloning for AI agents to understand message dialog and generate natural language communication strategies, the autonomous driving community currently lacks such datasets for vehicle-to-vehicle (V2V) natural language communication. Additionally, the modes of coordination between autonomous agents can be highly **diverse**, as supported by [2, 3], Figure 4, and videos from our supplementary materials. To address the challenges posed by this diversity and the absence of expert datasets, we integrated large language models (LLMs) throughout the decision-making and communication pipeline, incorporating LLM-based discussion and reflection to iteratively enhance the framework. Future work could further optimize the framework by leveraging collected datasets or decomposing the problem into smaller challenges, such as independently addressing message generation and message understanding.

---

> ### Author Response · Authors · 2024-11-22
> **General Response [2/3]**
>
> **Q2: Decentralized Problem Setting and General-Sum Game Formulation?**
> We formulated the problem using the language of multi-agent RL because the nature of the autonomous driving problem naturally aligns with a general-sum game structure [4]. Vehicles, whether controlled by humans or AI, exhibit diverse behaviors (e.g., risk-seeking or risk-averse) and **have goals and preferences that often conflict**. For instance, in our *Negotiation-Highway-Merge* scenario, both the vehicle on the highway and the one on the ramp aim to merge into the same lane, creating a conflict of intentions. One can conceptualize a payoff matrix per time step or game tree within this Markov game framework, where outcomes might include, i.e., time out if both are overly conservative, collision if both are aggressive and successful reconciliation if one vehicle adopts a polite strategy while the other is aggressive. To achieve both agents' goals, they must **resolve conflicts dynamically** during the interaction, often by compromising their initial preferences or time-value priorities. We deliberately avoided explicitly defining or optimizing a payoff matrix through gradient-based game-theoretic methods due to the sparsity of our reward structure, which was designed for evaluation clarity: +1 for successful completion, -1 for collisions, and 0 for being too conservative to complete the task. As highlighted in [5], formulating a detailed payoff matrix for every possible state in autonomous driving is highly complex and resource-intensive.
>
> With regards to **centralized control** of autonomous vehicles, although there are research studies on centralized control in driving [6], deploying such systems faces practical challenges. Centralized control requires transferring authority to a centralized system, which may not be feasible across car companies and may be unacceptable to many users, as vehicles are considered personal property. It also creates a single point of failure that can serve as an attack surface for malicious actors. In contrast, equipping individual vehicles with communication modules and AI assistants offers a more feasible solution that requires minimal infrastructure changes and preserves personal vehicle autonomy. Regarding smart city infrastructure, our problem setting does not prevent the synergies between vehicles and infrastructure; **infrastructure can be modeled as one of the agents in the system**.

---

> ### Author Response · Authors · 2024-11-22
> **General Response [3/3]**
>
> **Q3: Practicability of the Proposed Method?**
> We acknowledge that this work is a proof-of-concept exploratory study, and we fully recognize the challenges posed by the significant computational demands of foundation models, which currently hinder real-time application and restrict the scalability of the communication graphs. Addressing these limitations is a key focus of our future work. Potential solutions to improve the framework's inference speed include creating datasets for behavior cloning, applying RL self-play fine-tuning on smaller LLMs, exploring methodologies to internalize Chain-of-Thought (CoT) reasoning [9], and employing knowledge distillation to compress models into smaller architectures like GPT-2 [7] or even more compact models like [1], enabling deployment on mobile devices [8]. Notably, many prior works demonstrating the use of LLMs for driving also do not achieve real-time performance [10]. This study primarily lays a foundation for exploring natural language vehicle-to-vehicle (V2V) communication in autonomous driving, demonstrating that communication capabilities can be enhanced through iterative **interactions, discussions, and reflections**.
>
> Table 3 summarizes the average latencies and message sizes for each scenario under the communication setting, evaluated using Llama3-8B-Instruct on Nvidia A100 GPUs and Intel Gen 10 CPUs. The metrics include partial observable captioner latency (in seconds), reasoning latency (in seconds), decision latency (in seconds, excluding reasoning latency), and message size (in Mb). Data is aggregated over 10 episodes at each LLM decision step. Notably, GPT-4o online APIs demonstrate 2x faster generation speeds (\~16 seconds vs \~8 seconds). Scenarios without communication exhibit slightly lower reasoning and decision latencies compared to those with communication (\~16 seconds vs \~13 seconds), though the differences are within the same order of magnitude.
>
> **Table 3 Captioning, Reasoning, Decision Latency, Message Size using Llama3-8B-Instruct LLM Policy on Nvidia A100 GPUs**
>
> | | Perception Overtake | Perception Left Turn | Perception Red Light | Negotiation Overtake | Negotiation Highway Merge | Negotiation Highway Exit |
> | :--- | :---: | :---: | :---: | :---: | :---: | :---: |
> | Captioner Latency (s) | 0.022 | 0.023 | 0.025 | 0.022 | 0.017 | 0.016 |
> | Reasoning Latency (s) | 18.32 | 16.89 | 16.93 | 12.57 | 18.10 | 18.48 |
> | Decision Latency (s) | 2.83 | 2.25 | 2.37 | 1.56 | 1.57 | 1.60 |
> | Message Size (Mb) | 0.0016 | 0.0013 | 0.0014 | 0.0014 | 0.0005 | 0.0005 |
> |||||||
>
> ## References
> [1] Meta Fundamental AI Research Diplomacy Team (FAIR)†, Bakhtin A, Brown N, et al. Human-level play in the game of Diplomacy by combining language models with strategic reasoning[J]. Science, 2022, 378(6624): 1067-1074.
>
> [2] Strouse D J, McKee K, Botvinick M, et al. Collaborating with humans without human data[J]. Advances in Neural Information Processing Systems, 2021, 34: 14502-14515.
>
> [3] Cui, Brandon, et al. "Adversarial diversity in hanabi." The Eleventh International Conference on Learning Representations. 2023.
>
> [4] Dafoe, A., Hughes, E., Bachrach, Y., Collins, T., McKee, K. R., Leibo, J. Z., ... & Graepel, T. (2020). Open problems in cooperative AI. arXiv preprint arXiv:2012.08630.
>
> [5] Knox, W. B., Allievi, A., Banzhaf, H., Schmitt, F., & Stone, P. (2023). Reward (mis) design for autonomous driving. Artificial Intelligence, 316, 103829.
>
> [6] Guan, Y., Ren, Y., Li, S. E., Sun, Q., Luo, L., & Li, K. (2020). Centralized cooperation for connected and automated vehicles at intersections by proximal policy optimization. IEEE Transactions on Vehicular Technology, 69(11), 12597-12608.
>
> [7] Radford, A., Wu, J., Child, R., Luan, D., Amodei, D., & Sutskever, I. (2019). Language models are unsupervised multitask learners. OpenAI blog, 1(8), 9.
>
> [8] Liu, Z., Zhao, C., Iandola, F., Lai, C., Tian, Y., Fedorov, I., ... & Chandra, V. (2024). Mobilellm: Optimizing sub-billion parameter language models for on-device use cases. arXiv preprint arXiv:2402.14905.
>
> [9] Deng, Y., Choi, Y., & Shieber, S. (2024). From explicit cot to implicit cot: Learning to internalize cot step by step. arXiv preprint arXiv:2405.14838.
>
> [10] Wen, L., Fu, D., Li, X., Cai, X., Ma, T., Cai, P., ... & Qiao, Y. Dilu: A knowledge-driven approach to autonomous driving with large language models. The Twelfth International Conference on Learning Representations. 2024.

---

### Author Response · Authors · 2024-12-02
**Paper Revision Notes**

Dear reviewers,

Thank you for your valuable feedback. We have revised the paper based on our discussions, with changes highlighted in blue in the latest PDF file. Below is a summary of the key updates:
- Enhanced clarity on key concepts throughout the paper:
  - Motion controls are executed through high-level commands controlled by our method and implemented by low-level path planners and PID controllers.
  - Scoping our main contributions:
      - A novel simulation framework for V2V in natural language, allowing heterogeneous agent configuration and scenarios of interest;
      - An agent framework to enable V2V communication in natural language with a novel learning methodology realized by multi-agent LLM agent discussion.
  - Highlight the mechanism of the partially observable perception module **[L278-282]**;
  - Clearer methodology description **[L349-368]**;
- Extended discussion on limitations and future works **[Appendix E]**;
- Updated additional result table reflecting an improved partial observable captioner **[Appendix D]**;
  - The improvement in the environment description mainly optimizes the description of roads and lanes with semantic meanings instead of just road ID or lane ID.
- More Qualitative Examples of knowledge learned by the debriefing method **[Appendix F]**

We sincerely appreciate the reviewers' thoughtful feedback and kindly request that the evaluation focus on the contributions explicitly outlined in our submission. This work aims to explore natural language communication for V2V driving and an LLM-based multi-agent learning framework. We acknowledge that our methodology is not yet ready for deployment on roads and recognize concerns such as latency issues brought about by LLM reasoning, communication system-specific challenges, and potential improvements in learning stabilities. While these concerns cannot be fully addressed within the rebuttal phase, we view them as important directions for future work.

Thank you for your engagement and feedback on this work.

Sincerely and respectfully,

The Authors

---

### Public Comment · ~Jiaxun_Cui1 · 2025-05-30
**BibTex**

If you find our paper useful to your research, please use the following BibTex instead of the OpenReview one:
```
@misc{cui2025talkingvehicles,
      title={Towards Natural Language Communication for Cooperative Autonomous Driving via Self-Play},
      author={Jiaxun Cui and Chen Tang and Jarrett Holtz and Janice Nguyen and Alessandro G. Allievi and Hang Qiu and Peter Stone},
      year={2025},
      eprint={2505.18334},
      archivePrefix={arXiv},
      primaryClass={cs.RO},
      url={https://arxiv.org/abs/2505.18334},
}
```

---

### Meta-Review · Area_Chair_Hgbh · 2024-12-20

**Metareview:**

The paper proposes using natural language for communication between autonomous vehicles in traffic scenarios. It introduces LLM+Debrief, a method where vehicles use LLMs to generate messages and actions, and then engage in "debriefing" sessions to improve performance. It is evaluated in a simulated environment built on CARLA.

Strengths:
Novel approach to vehicle-to-vehicle communication.
Uses a realistic simulator (CARLA).
Includes debriefing sessions for improved learning.

Weaknesses:
Unclear why natural language is necessary for communication.
Limited evaluation and lack of real-time considerations.
Assumes cooperative vehicles, which is unrealistic.
Performance is worse than some baselines.
Concerns about scalability and message congestion.

Based on the scores (3, 3, 3, 6), the paper does not meet the acceptance bar, and I wish them best of luck for revisions and resubmission of their work at another venue.

**Additional Comments On Reviewer Discussion:**

The authors did extensively engage with the reviewers through ample rebuttals, but did not manage to address the important points in the discussions, namely on the necessity of using natural language for communication, handling congestion, and adversarial information.

---

### Decision · Program_Chairs · 2025-01-22

Reject